# Boundary Decomposition for Nadir Objective Vector Estimation

**Ruihao Zheng**   **Zhenkun Wang**[*]
School of System Design and Intelligent Manufacturing,
Southern University of Science and Technology
12132686@mail.sustech.edu.cn, wangzhenkun90@gmail.com

## Abstract

The nadir objective vector plays a key role in solving multi-objective optimization problems (MOPs), where it is often used to normalize the objective space and guide the search. The current methods for estimating the nadir objective vector perform effectively only on specific MOPs. This paper reveals the limitations of these methods: exact methods can only work on discrete MOPs, while heuristic methods cannot deal with the MOP with a complicated feasible objective region. To fill this gap, we propose a general and rigorous method, namely boundary decomposition for nadir objective vector estimation (BDNE). BDNE scalarizes the MOP into a set of boundary subproblems. By utilizing bilevel optimization, boundary subproblems are optimized and adjusted alternately, thereby refining their optimal solutions to align with the nadir objective vector. We prove that the bilevel optimization identifies the nadir objective vector under mild conditions. We compare BDNE with existing methods on various black-box MOPs. The results conform to the theoretical analysis and show the significant potential of BDNE for real-world application.

## 1 Introduction

The multi-objective optimization problem (MOP) can be written as

$$\begin{aligned}
\text{min.} \quad & \mathbf{f}(\mathbf{x}) = (f_1(\mathbf{x}), \ldots, f_m(\mathbf{x}))^\intercal, \\
\text{s.t.} \quad & \mathbf{x} \in \Omega,
\end{aligned} \tag{1}$$

where $\mathbf{x} = (x_1, \ldots, x_n)^\intercal$ is the decision vector (also called solution), and $\Omega \subset \mathbb{R}^n$ denotes the feasible region. $\mathbf{f} : \mathbb{R}^n \to \mathbb{R}^m$ is composed of $m$ objective functions, and $\mathbf{f}(\mathbf{x})$ is the objective vector corresponding to $\mathbf{x}$.

**Definition 1.** *Given two vectors $\mathbf{u}, \mathbf{v} \in \mathbb{R}^m$, $\mathbf{u}$ is said to **dominate** $\mathbf{v}$ (denoted as $\mathbf{u} \prec \mathbf{v}$), if and only if $u_i \leq v_i$ for every $i \in \{1, \ldots, m\}$ and $u_j < v_j$ for at least one $j \in \{1, \ldots, m\}$.*

**Definition 2.** *A decision vector $\mathbf{x}^*$ and the corresponding objective vector $\mathbf{f}(\mathbf{x}^*)$ are **Pareto-optimal**, if there is no $\mathbf{x} \in \Omega$ such that $\mathbf{f}(\mathbf{x})$ dominates $\mathbf{f}(\mathbf{x}^*)$ according to Definition 1.*

**Definition 3.** *A decision vector $\mathbf{x}^* \in \Omega$ and the corresponding objective vector $\mathbf{f}(\mathbf{x}^*)$ are **weakly Pareto-optimal**, if there does not exist another decision vector $\mathbf{x} \in \Omega$ such that $f_i(\mathbf{x}) < f_i(\mathbf{x}^*)$ for all $i = 1, \ldots, m$.*

**Definition 4.** *The set of all Pareto-optimal solutions is called the **Pareto set** (denoted as $PS$), and the set of all Pareto-optimal objective vectors is called the **Pareto front** (denoted as $PF$).*

**Definition 5.** *The relative complement of the $PF$ in the set of all weakly Pareto-optimal objective vectors is called the **weakly Pareto-optimal boundary** (denoted as $WPB$).*

---

[*]Corresponding author

38th Conference on Neural Information Processing Systems (NeurIPS 2024).

**Definition 6.** *The **ideal objective vector** $\mathbf{z}^{ide}$ is composed of the lower bounds of the $PF$, i.e., $z_i^{ide} = \min_{\mathbf{x} \in PS} f_i(\mathbf{x})$ for $i = 1, \ldots, m$. The **nadir objective vector** $\mathbf{z}^{nad}$ consists of the upper bounds of the $PF$, i.e., $z_i^{nad} = \max_{\mathbf{x} \in PS} f_i(\mathbf{x})$ for $i = 1, \ldots, m$.*

**Definition 7.** *For the $i$-th objective function $f_i$ with $i \in \{1, \ldots, m\}$, an objective vector $\mathbf{z}^{(i)^c}$ is called the **critical point** of $f_i$ if it is Pareto-optimal and has the worst $f_i$ value, i.e., $\mathbf{z}^{(i)^c} \in \{\mathbf{f}(\mathbf{x}) | \mathbf{x} = \arg\max_{\mathbf{x} \in PS} f_i(\mathbf{x})\}$. The nadir objective vector is often determined by obtaining the critical point on each objective, since $z_i^{nad} = z_i^{(i)^c}$ for each $i \in \{1, \ldots, m\}$.*

The nadir objective vector is a fundamental concept of multi-objective optimization and has been widely applied to real-world problems [1, 2, 3]. Specifically, numerous optimization methods' operations involve the nadir objective vector. Firstly, the nadir objective vector is often used to guide the search in various tasks, including both continuous MOPs [4, 5] and discrete MOPs [6, 7]. An inaccurate estimation of the nadir objective vector degrades the performance of exact algorithms [8], evolutionary algorithms [9], and multi-objective learning [10]. Secondly, the nadir objective vector offers a comprehensive view of the $PF$ for decision-makers, which can facilitate the application of preference-based algorithms [11, 12, 2]. For example, an accurate nadir objective vector is the assumption of many interactive algorithms. A poor approximation may cause biased decisions. In addition, the nadir objective vector and the ideal objective vector are widely used to normalize the objective space [13]. Normalization with an inappropriately estimated nadir objective vector can cause performance deterioration of the optimization algorithm [14, 15].

The ideal objective vector can be acquired by minimizing each objective separately. Unfortunately, obtaining the nadir objective vector is much more complicated [16, 17]. Existing exact methods are developed for discrete MOPs, posing significant challenges for their extension to other MOPs. The remaining methods are heuristic, showing satisfactory performance on simple MOPs. For example, DTLZ1 has an equilateral-triangle-shaped $PF$, and its nadir objective vector can be easily estimated using several heuristic methods [9]. However, when the MOP possesses a complicated feasible objective region (*e.g.*, an irregular $PF$ and the $WPB$), all heuristic methods fail to approximate the nadir objective vector accurately.

In this paper, we first demonstrate the shortcomings of existing methods in detail. After that, we propose a general method with theoretical guarantees called boundary decomposition for nadir objective vector estimation (BDNE). We implement BDNE for black-box MOPs and use 28 black-box problems to validate its performance. The results indicate that BDNE remarkably outperforms the existing methods. The major contributions of BDNE are summarized as follows:

- Scalarization method for boundary decomposition. We define the boundary subproblem that converts an objective vector into a scalar via boundary weight vectors. We prove that the critical point of each objective can be found by optimizing a particular boundary subproblem under mild conditions (*i.e.*, the critical point satisfies proper Pareto optimality). We also prove that the optimal solution to any boundary subproblem is Pareto-optimal, thereby facilitating the search of the particular boundary subproblems.

- Bilevel optimization based on boundary decomposition. We formulate a bilevel optimization problem for each objective, aiming to identify the particular boundary subproblem by comparing the optimal solutions of boundary subproblems. The upper-level optimization seeks each objective's boundary weight vector that maximizes the objective function value; the lower-level optimization searches for the optimal solutions of given boundary subproblems. Besides, the trade-off of decision-makers can be involved in finding a satisfactory nadir objective vector.

## 2  Related Works

Existing nadir objective vector estimation methods can be divided into two categories: 1) exact methods and 2) heuristic methods.

**Exact Method.** Exact methods are all designed for discrete MOPs and cannot be applied to other problems. For example, the methods in [18, 19] assume that the objective function values are integers, limiting their applicability to continuous problems. Moreover, some exact methods are exclusively designed for multi-objective integer linear programming [20]. $m$ bilevel optimization problems are

formulated in [21] and guarantee that their optimal solutions construct the nadir objective vector. The bilevel optimization is implemented solely for the discrete search space where an exhaustive search is available. However, the exact solver is often unavailable for many problems such as non-linear continuous ones, leading to uncertain optimality gaps. As a result, the pay-off table, which this method relies heavily on, may be tough to obtain. In addition, its lower-level optimization, including two stages, may exhibit significant unreliability and high computational costs. Another significant limitation of exact methods is their inability to solve beyond small-scale problems within reasonable runtimes, thus posing substantial challenges to their real-world applicability.

**Heuristic Method.** Heuristic methods readily apply to diverse MOPs yet lack theoretical guarantees. Generally, the heuristic method estimates the nadir objective vector as follows

$$\hat{z}_i^{nad} = \max_{\mathbf{x} \in S} f_i(\mathbf{x}) \text{ for } i = 1, \dots, m, \tag{2}$$

where $S$ is an iteratively improved solution set. $S$ can be specified as the population of the multi-objective evolutionary algorithm (MOEA) [22] (denoted as SF1). When the population is the $PS$, Eq. (2) becomes the definition of the nadir objective vector. That is, SF1 can achieve the nadir objective vector under the ideal situation where the population is the $PS$. However, this is almost impossible because retaining inferior solutions, such as weakly Pareto-optimal or dominance-resistant ones, in the population is often inevitable [23]. In case the population contains these inferior solutions, the SF1-based algorithm may be severely misled and fail to estimate the nadir objective vector, as shown in Figure 2b and Figure 2f (see Section 4). Several methods are proposed to mitigate the impact of inferior solutions by selecting a subset from the population [24, 25, 26]. Additionally, $S$ can be obtained by identifying extreme points in the population. In [27, 28, 29], the extreme points are determined by finding the objective vector with minimum values on each objective function. This method is also known as the pay-off table method, which can overestimate or underestimate the nadir objective vector [30, 20]. Some methods identify the objective vectors that are closest to axis vectors as extreme points. The distance between the objective vector and the axis vector can be measured by the Minkowski distance (*e.g.*, $L_2$ [31] and $L_\infty$ [32]), the perpendicular distance [33], and the cosine similarity [9]. In [34], each objective is associated with a single-objective optimization subproblem to determine $m$ extreme points. In Appendix A, we use an example to illustrate the deficiencies of heuristic methods in estimating the nadir objective vector. We can observe from Figure 4 and Table 4 that all methods incorrectly estimate the critical points as well as the nadir objective vector.

## 3 Methodology

To address the limited applicability of exact methods and the unreliability of heuristic methods, we propose a method with general applicability to any MOP and theoretical guarantees in this section. Specifically, our method does not necessitate the objective function value to be an integer as opposed to [18, 19] and involves simpler optimization tasks compared to [21]. Our method also enables finding the nadir objective vector, unlike heuristic methods. Furthermore, our method supports using a user-defined trade-off. In the following, we begin by introducing boundary decomposition and establishing its theoretical foundations. After that, we develop a bilevel optimization method based on boundary decomposition to achieve alignment with the nadir objective vector. This method is then implemented for the black-box MOP, and its effectiveness is evaluated in the subsequent section.

### 3.1 Boundary Subproblem

We define a boundary subproblem for the $i$-th objective as

$$g_i^{bd}(\mathbf{x}|\mathbf{w}^i, \mathbf{z}^r, \alpha) = \max_{1 \le j \le m} \left\{ w_j^i \left( (1-\alpha)f_j(\mathbf{x}) + \frac{\alpha}{m} \sum_{k=1}^m f_k(\mathbf{x}) - z_j^r \right) \right\}, \tag{3}$$

where $\mathbf{w}^i$ is called the boundary weight vector of the $i$-th objective, $\mathbf{z}^r$ is a reference point, and $0 < \alpha < 1$. $\mathbf{w}^i$ satisfies three conditions: 1) $w_i^i = 0$; 2) $\forall j \in \{1, \dots, m\}, w_j^i \ge 0$; 3) $\exists j \in \{1, \dots, m\}, w_j^i > 0$. The definition of the boundary subproblem is inspired by the modified weighted Tchebycheff metric [35]. The contour surface of the boundary subproblem is illustrated in Figure 1. Additionally, several examples are presented to demonstrate the optimal objective vector of the boundary subproblem. In the following, we provide the theoretical foundations. Claims are followed by corresponding explanations. All proofs are presented in Appendix F.

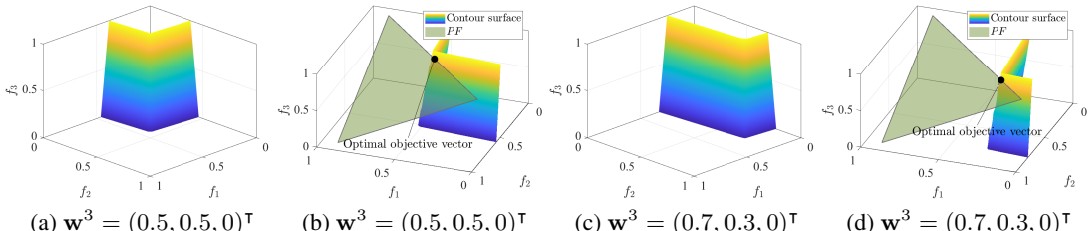

(a) $\mathbf{w}^3 = (0.5, 0.5, 0)^\intercal$     (b) $\mathbf{w}^3 = (0.5, 0.5, 0)^\intercal$     (c) $\mathbf{w}^3 = (0.7, 0.3, 0)^\intercal$     (d) $\mathbf{w}^3 = (0.7, 0.3, 0)^\intercal$

Figure 1: The contour surfaces of boundary subproblems.

**Definition 8** (From [36]). *Let $\tilde{\mathbf{z}}$ be the image of an objective vector $\mathbf{z}$, where*

$$\tilde{z}_i = (1 - \alpha)z_i + \frac{\alpha}{m}\sum_{j=1}^{m} z_j \ \text{for} \ i = 1, \ldots, m \ \ (0 < \alpha < 1). \tag{4}$$

*Given a value of $\alpha$ and two objective vectors $\mathbf{u}$ and $\mathbf{v}$, $\mathbf{u}$ is said to **cone-dominate** $\mathbf{v}$ (denoted as $\mathbf{u} \prec^c \mathbf{v}$) if and only if $\tilde{\mathbf{u}} \prec \tilde{\mathbf{v}}$.*

**Definition 9.** *Given a value of $\alpha$, an objective vector is cone-optimal if no objective vector can cone-dominate it.*

**Theorem 1.** *The optimal objective vector to the boundary subproblem with $\mathbf{w}^i$ (denoted as $\mathbf{z}^*$) must be Pareto-optimal, if and only if $\exists j \in \{1, \ldots, m\} \setminus \{i\}$ such that $z_j^r \leq \tilde{z}_j^*$.*

**Theorem 2.** *Let $z_j^r \leq \tilde{z}_j^{(i)^c}$ for every $j \in \{1, \ldots, m\} \setminus \{i\}$. $\mathbf{z}^{(i)^c}$ uniquely solves the boundary subproblem with*

$$w_j^i = \begin{cases} 0, & j = i, \\ \dfrac{\beta}{\tilde{z}_j^{(i)^c} - z_j^r}, & j \neq i, \end{cases} \tag{5}$$

*if and only if $\mathbf{z}^{(i)^c}$ is cone-optimal.*

**Corollary 1.** *To ensure the validity of Theorem 2 without further information, it is necessary to satisfy the condition $z_j^r \leq \tilde{z}_j^{(i)^c}$ for every $j \in \{1, \ldots, m\} \setminus \{i\}$.*

Firstly, we reveal the connection between the boundary subproblem and the nadir objective vector. The boundary subproblem involves two steps: transforming the given objective vector and scalarizing the transformed objective vector with the boundary weight vector. The transformation implies cone domination, which defines a strict partial order (see Appendix F.1 for details) and belongs to the family of generalized Pareto domination [37]. We assume that the critical points are cone-optimal and a set of objective vectors is obtained by optimizing infinitely sampling boundary subproblems. According to Theorems 1 and 2, this set is a subset of the $PF$ and the critical points are included in it. Finally, the nadir objective vector can be identified by Eq. (2) where $S$ is specified as the obtained set.

**Definition 10** (From [38]). *A decision vector $\mathbf{x}^*$ and the corresponding objective vector $\mathbf{f}(\mathbf{x}^*)$ are **properly Pareto-optimal** if they are Pareto-optimal and if there exists a finite number $M > 0$ such that, for each $i$ and any $\mathbf{x} \in \Omega$, we have*

$$\frac{f_i(\mathbf{x}^*) - f_i(\mathbf{x})}{f_j(\mathbf{x}) - f_j(\mathbf{x}^*)} \leq M, \tag{6}$$

*where $j$ satisfies $f_j(\mathbf{x}^*) < f_j(\mathbf{x})$.*

**Theorem 3.** *An objective vector is cone-optimal, if and only if the objective vector is Pareto-optimal and the value of $\alpha$ satisfies*

$$\alpha \leq \frac{m}{(m-1)M + m}. \tag{7}$$

**Corollary 2.** *A cone-optimal objective vector is also properly Pareto-optimal.*

The following question arises: under what conditions can a solution be considered cone-optimal? Let $\mathbf{u} = (0,1,1)^\intercal$ and $\mathbf{v} = (0.2, 0.5, 0.5)^\intercal$ be a critical point and Pareto-optimal objective vector, respectively. If $\alpha = 0.5$, then $\mathbf{v} \prec^c \mathbf{u}$ (since $\tilde{\mathbf{u}} = (1,2,2)^\intercal$ and $\tilde{\mathbf{v}} = (0.8, 1.1, 1.1)^\intercal$); If $\alpha = 0.1$, then $\mathbf{u} \not\prec^c \mathbf{v}$ and $\mathbf{v} \not\prec^c \mathbf{u}$ (since $\tilde{\mathbf{u}} = (0.2, 1.2, 1.2)^\intercal$ and $\tilde{\mathbf{v}} = (0.32, 0.62, 0.62)^\intercal$). The critical point is cone-dominated if an excessive $\alpha$ value is used. We should know how to set an appropriate $\alpha$ to preserve the specific Pareto-optimal objective vectors. We find that cone domination is related to proper Pareto optimality. The idea of proper Pareto optimality is to divide the $PS$ into proper and improper ones. The definition of proper Pareto optimality is shown in Definition 10. That is, a solution can be considered properly Pareto-optimal when at least one pair of objectives satisfies: a finite decrement in one objective requires a reasonable increment in the other objective. We also can infer that different proper Pareto-optimal solutions may necessitate distinct minimum values of $M$. Then we derive that $\alpha$ is bounded by $M$ as shown in Theorem 3. Corollary 2 is established accordingly. We can let $\alpha$ be a sufficiently small value such that the critical points are cone-optimal.

**Theorem 4.** *Let $\mathbf{z}^*$ be an objective vector satisfying $z_j^r \leq \tilde{z}_j^*$ for every $j \in \{1, \ldots, m\} \setminus \{i\}$. If an objective vector $\mathbf{z}'$ does not cone-dominate $\mathbf{z}^*$ and $z_i^* \geq z_i'$, then $\mathbf{z}^*$ has a lower function value than $\mathbf{z}'$ with respect to some boundary subproblem of the $i$-th objective.*

**Corollary 3.** *Let $\mathbf{z}^e$ consist of the optimal values of $m$ BLOPs and $\{\mathbf{z} | \mathbf{z} \prec \mathbf{z}^e\}$ be the promising region. Any Pareto-optimal objective vector outside the promising region fails to address the trade-off of decision-makers, namely, $M > \mu$.*

The remaining issue is that $M$ is not known in advance, making it difficult to set $\alpha$ according to $M$. We introduce a user-defined parameter $\mu$ ($\mu > 0$) and let $\alpha = \frac{m}{(m-1)\mu + m}$. $\mu$ represents the upper bound of the trade-off between the $k$-th and $l$-th objectives, where $k = \arg\max_{1 \leq j \leq m} f_j(\mathbf{x}^*) - f_j(\mathbf{x})$ and $l = \arg\max_{1 \leq j \leq m} f_j(\mathbf{x}) - f_j(\mathbf{x}^*)$. In other words, $\mu$ is the amount of increment in the value of one objective function that the decision-maker is willing to tolerate in exchange for a one-unit decrement in another objective function. If the preference of decision-makers about $\mu$ is available, we have the guarantee shown in Corollary 3 derived from Theorems 3 and 4. Corollary 3 can guarantee to obtain a satisfactory nadir objective vector. When the preference is not provided, $\mu$ can take a sufficiently large value. On the one hand, if a finite trade-off $M$ exists for the critical point, then $\mu$ can take an appropriate value from $[M, \infty)$ to obtain the nadir objective vector. On the other hand, $M \to \infty$ results in $\alpha \to 0$, which means the corresponding objective vector is not properly Pareto-optimal. In practical terms, a Pareto-optimal solution with a very large value of $M$ does not essentially differ from an inferior solution for decision-makers. If the critical points are improper, using a generally agreeable value of $\mu$ is reasonable.

**Theorem 5.** *The optimal objective vector to the boundary subproblem with $\mathbf{w}^i$ (denoted as $\mathbf{z}^*$) must be cone-optimal (or properly Pareto-optimal), if $\exists j \in \{1, \ldots, m\} \setminus \{i\}$ such that $z_j^r < \tilde{z}_j^*$.*

**Theorem 6.** *Let $z_j^r < \tilde{z}_j^*$ for every $j \in \{1, \ldots, m\} \setminus \{i\}$. If an objective vector $\mathbf{z}^*$ is optimal for the boundary subproblem with $\mathbf{w}^i$, $\mathbf{z}^*$ must be an optimal objective vector to the boundary subproblem with*

$$w_j^* = \begin{cases} 0, & j = i, \\ \dfrac{\beta}{\tilde{z}_j^* - z_j^r}, & j \neq i. \end{cases} \tag{8}$$

Moreover, we can make minor modifications to Theorem 1, resulting in Theorem 5. Theorem 6 can be deduced using Theorem 5. Theorem 6 reveals the relationship between the boundary subproblem's optimal solution and the boundary weight vector, which is useful for designing the algorithm in Section 3.3.

## 3.2 Nadir Objective Vector Estimation via Bilevel Optimization

Practically, we cannot have infinite boundary subproblems. In this subsection, an optimization problem is defined to obtain the nadir objective vector. Its two goals are: 1) converging to Pareto-optimal objective vectors and 2) identifying critical points from Pareto-optimal ones. In the 2-objective case, both goals can be achieved through a single optimization procedure. Specifically, the critical points can be obtained by optimizing boundary subproblems using $(1,0)^\intercal$ and $(0,1)^\intercal$ as the weight vectors. This is because solving the boundary subproblem with $\mathbf{w}^i$ is equivalent to solving that with $\beta\mathbf{w}^i$ where $\beta > 0$ is a constant. Unfortunately, the critical points cannot be obtained by a

fixed boundary subproblem when the MOP involves three or more objectives. That is, both goals cannot be achieved simultaneously. This motivates the formulation of a bilevel optimization problem (BLOP) for each objective to achieve the two goals: the lower-level optimization problem (LLOP) corresponds to the first goal while the upper-level optimization problem (ULOP) is for the second goal. The BLOP is formulated below, based on the boundary subproblem.

Firstly, $M$ is affected by different value ranges of objective functions, which might lead to algorithm performance deterioration [39]. The objective space should be normalized properly. We let $z_j^r = 0$ for $j = 1, \ldots, m$ in Eq. (3) and introduce two reference points $\mathbf{z}^{r1}$ and $\mathbf{z}^{r2}$ ($z_j^{r2} > z_j^{r1}$ for $j = 1, \ldots, m$) for normalization. Then, the boundary subproblem with normalization can be written as

$$g_i^{bdn}(\mathbf{x}|\mathbf{w}^i, \mathbf{z}^{r1}, \mathbf{z}^{r2}, \alpha) = \max_{1 \leq j \leq m} \left\{ w_j^i \left( (1-\alpha)f_j'(\mathbf{x}) + \frac{\alpha}{m} \sum_{k=1}^m f_k'(\mathbf{x}) \right) \right\}, \qquad (9)$$

where $f_k'(\mathbf{x}) = (f_k(\mathbf{x}) - z_k^{r1})/\left(z_k^{r2} - z_k^{r1}\right)$ for $k = 1, \ldots, m$. In this formulation, $\mathbf{z}^{r1}$ should be set according to Theorem 1 and Corollary 1. For example, it is reasonable to set $\mathbf{z}^{r1}$ to $\mathbf{z}^{ide}$ since $z_j^{ide} \leq f_j(\mathbf{x})$ for $j = 1, \ldots, m$ such that $(1-\alpha)f_j'(\mathbf{x}) + \frac{\alpha}{m}\sum_{k=1}^m f_k'(\mathbf{x}) \geq 0$. Furthermore, $\mathbf{z}^{r2}$ should be set according to [40]. Secondly, several boundary subproblems might have the same optimal solution. On the one hand, solving the boundary subproblem with $\mathbf{w}^i$ is equivalent to solving that with $\beta \mathbf{w}^i$ where $\beta > 0$ is a constant. We can let $\sum_{j=1 \wedge j \neq i}^m w_j^i = 1$. On the other hand, Theorem 6 indicates that the optimal objective vectors to some boundary subproblems may not align with the corresponding boundary weight vectors. For example, this scenario is common on discrete MOPs. We can penalize the flat landscape of the ULOP according to the distance between the boundary weight vector and its optimal objective vector. Let $\mathbf{x}^*$ be the optimal solution of the boundary subproblem with $\mathbf{w}^i$. In this paper, we calculate the distance as $d(\mathbf{w}^i, \mathbf{x}^*) = \sqrt{\sum_{j=1 \wedge j \neq i}^m \left( w_j^i - u_j / \sum_{k=1}^m u_k \right)^2}$ where

$$u_j = \begin{cases} 0, & j = i, \\ \dfrac{1}{(1-\alpha)f_j'(\mathbf{x}) + \frac{\alpha}{m}\sum_{k=1}^m f_k'(\mathbf{x})}, & j \neq i. \end{cases} \qquad (10)$$

Let $\epsilon$ be a sufficiently small constant, $l \in \{1, \ldots, m\} \setminus \{i\}$, and $I = \{1, \ldots, m\} \setminus \{i, l\}$. Finally, the BLOP with respect to the $i$-th objective for $i = 1, \ldots, m$ has the ULOP formulated as

$$\begin{aligned} \text{max.} \quad & f_i^u(\mathbf{w}^i) = f_i(\mathbf{x}^*) - \epsilon d(\mathbf{w}^i, \mathbf{x}^*), \\ \text{s.t.} \quad & w_i^i = 0, \quad w_l^i = 1 - \sum_{j \in I} w_j^i, \quad \sum_{j \in I} w_j^i \leq 1, \\ & 0 \leq w_j^i \leq 1 \text{ for every } j \in I, \end{aligned} \qquad (11)$$

where $\mathbf{x}^*$ is the optimal decision vector to the LLOP of the following form

$$\begin{aligned} \text{min.} \quad & f_i^l(\mathbf{x}) = g_i^{bdn}(\mathbf{x}|\mathbf{w}^i, \mathbf{z}^{r1}, \mathbf{z}^{r2}, \alpha), \\ \text{s.t.} \quad & \mathbf{x} \in \Omega. \end{aligned} \qquad (12)$$

The search is performed on the original decision space in the LLOP, while the search space of the ULOP can be viewed as $\mathbb{R}^{m-2}$ (*i.e.*, $(m-2)$-dimensional Euclidean space). For the $i$-th objective, the ULOP uses a boundary weight vector $\mathbf{w}^i$ as a solution and requires maximizing the $i$-th penalized objective function value of the Pareto-optimal solution obtained by the LLOP. In other words, a feasible solution of the ULOP requires an optimal solution of the LLOP. We propose an algorithm framework called BDNE for estimating the nadir objective vector on the MOP with more than two objectives. Specifically, BDNE aims to solve these $m$ BLOPs. The steps of BDNE are given in Algorithm 1, where $\mu$ is the user-defined upper bound of the trade-off (see Section 3.1 for details). Three following issues should be specified: determine the reference points $\mathbf{z}^{r1}$ and $\mathbf{z}^{r2}$, choose suitable single-objective optimizers, and set the stopping criteria.

### 3.3 Algorihm for Black-Box Multi-Objective Optimization Problems

To evaluate the viability of BDNE, we implement it for black-box MOPs. We adopt evolutionary algorithms as the solvers in BDNE. Each ULOP uses CMA-ES [41] to search the eligible boundary

---
**Algorithm 1** BDNE
---
**Input**: An MOP, stopping criteria of $m$ BLOPs, $\mu$
**Output**: $\hat{\mathbf{z}}^{nad}$
1: Initialization: $\alpha \leftarrow \frac{m}{(m-1)\mu+m}$; determine $\mathbf{z}^{r1}$ and $\mathbf{z}^{r2}$; configure solvers.
2: **while** stopping criteria of $m$ ULOPs are not all satisfied **do**
3:  Generate boundary weight vectors by the solvers of unstopped ULOPs.
4:  Minimize LLOPs with the generated boundary weight vectors by the corresponding solvers.
5:  Update parameters: $\mathbf{z}^{r1}$ and $\mathbf{z}^{r2}$ (optional); parameters in the solvers of unstopped ULOPs.
6: **end while**
---

weight vector(s). We utilize CMA-ES to optimize the ULOP for two reasons. First, CMA-ES is a state-of-the-art algorithm for single-objective black-box optimization. Second, optimal solutions to the LLOPs are not always available; instead, approximate solutions are often obtained. Consequently, the function values of the approximate solutions may exhibit noise. CMA-ES is suitable for this task as it demonstrates strong robustness in optimizing noisy functions [41]. In each iteration, $m$ ULOPs generate the pending LLOPs simultaneously. Then all LLOPs are solved collaboratively by the MOEA instead of optimizing each LLOP separately. This is because the superiority of MOEAs is demonstrated empirically and theoretically in solving multi-objective black-box optimization problems [42, 43].

Table 1: Notation used in Section 3.3.

| Symbol | Description |
|---|---|
| $\tau_u(\tau_l)$ | Maximum number of iterations for each CMA-ES procedure (the MOEA solving LLOPs). |
| $N$ | $\frac{N}{m} \in \mathbb{Z}$; number of generated LLOPs in each iteration of $m$ ULOPs. |
| $P$ | $|P| = N$; population of the MOEA for solving $N$ LLOPs. |
| $A$ | $|A| = N$; elite archive preserved the current best solutions for $N$ LLOPs. |
| $\mathbf{w}^{i,j}$ | The $j$-th boundary weight vector of the $i$-th objective. |

**Stopping Criteria and Reference Points.** The maximal number of iterations is employed as the stopping criterion for the ULOP as well as the LLOP. We use $\tau_u$ and $\tau_l$ to denote the maximum number of iterations for the ULOP and the LLOP respectively. $\mathbf{z}^{r1}$ is set to the current best objective function values. $\mathbf{z}^{r2}$ is constructed by the current best objective function values of $m$ ULOPs. Initially, EC-NSGA-II [22] runs for $\tau_l$ iterations. The minimum and maximum objective function values of its final population determine the settings of $\mathbf{z}^{r1}$ and $\mathbf{z}^{r2}$. Then $\mathbf{z}^{r1}$ is updated once a new solution is generated. $\mathbf{z}^{r2}$ is only renewed when the parameters of CMA-ES update. During the optimization process, $\mathbf{z}^{r1}$ and $\mathbf{z}^{r2}$ iteratively approximate $\mathbf{z}^{ide}$ and $\mathbf{z}^{nad}$ respectively.

**Upper-Level Optimization.** $m$ ULOPs represents $m$ CMA-ES procedures. Let $\iota = \frac{N}{m} - 1$. The population size of each CMA-ES procedure is $\iota$. That is, $(N - m)$ boundary weight vectors are used to align with the eligible ones and evenly assigned to $m$ objectives. Let $\{\mathbf{w}^{i,j}, j = 1, \dots, \iota\}$ denote the set of adjustable boundary weight vectors for the $i$-th objective. Initially, most of them are sampled from the initial distribution of the CMA-ES procedure. The sampled points may need to be repaired to satisfy the constraints in the ULOP. For each objective, one boundary weight vector is initialized to the vector with $(m - 1)$ elements being $\frac{1}{m-1}$ (e.g., $\left(0, \frac{1}{2}, \frac{1}{2}\right)^\mathsf{T}$). The remaining $m$ ones, which are $m$ different unit vectors, are fixed throughout the optimization of $m$ BLOPs. The LLOPs with these $m$ boundary weight vectors can motivate the search of $\mathbf{z}^{ide}$. Besides, they enhance population diversity when the adjustable boundary subproblems become similar. When LLOPs are stopped, the parameters of each CMA-ES procedure are updated first according to the $\iota$ boundary weight vectors. Let $\mathbf{x}^{i,j} \in A$ be the best solution so far for the subproblem with $\mathbf{w}^{i,j}$, and then the fitness of $\mathbf{w}^{i,j}$ is defined as $\left(f_i\left(\mathbf{x}^{i,j}\right) - \epsilon d\left(\mathbf{w}^{i,j}, \mathbf{x}^{i,j}\right)\right)$. Note that some boundary weight vectors, such as repaired ones, might not be directly generated from the current distribution of the CMA-ES procedure. These injected solutions should obey the injection rule [44, 45]. Subsequently, the boundary weight vectors of the $i$-th objective are updated as follows:

**Step 1** $W \leftarrow \{\mathbf{w}^{i,j}, j = 1, \dots, \iota\}$ and then delete $\lfloor \frac{\iota}{2} \rfloor$ the worst boundary weight vector in $W$.

**Step 2** Transform the best one in $W$ according to Eq. (10).

**Step 3** Create $\lfloor \frac{\iota}{2} \rfloor$ new ones via the $i$-th CMA-ES procedure and integrate them into $W$.

The updated $\mathbf{z}^{r2}$ may substantially alter the normalized space in Eq. (9), potentially changing the optimal solution to the LLOP. As Theorem 6 reports, Step 2 can effectively preserve the best solution for the LLOP, irrespective of changes in the normalized space. Particularly, in the last iteration of the ULOP (*i.e.*, the $\tau_u$-th iteration), $\mathbf{w}^{i,1}, \ldots, \mathbf{w}^{i,\iota}$ are all changed to the boundary weight vector with the best fitness for $i = 1, \ldots, m$. More computational resources are allocated to the $i$-th objective's current best LLOP in the final iteration. It helps to obtain a better approximation for these $m$ LLOPs.

**Lower-Level Optimization.** $N$ LLOPs are optimized by an MOEA with population size $N$. An evolutionary algorithm executes solution reproduction and environmental selection iteratively. The reproduction procedure of the proposed MOEA includes two steps:

**Step 1** Use binary tournament selection based on $\boldsymbol{\pi}$ to obtain the mating pool from $P$.

**Step 2** Apply reproduction operators to create an offspring set of size $N$ (denoted as $O$).

$\boldsymbol{\pi}$ is a utility vector including utility values of solutions in $P$. A solution with a lower utility value is more likely to enter the mating pool. The selection procedure is performed as follows:

**Step 1** Calculate $\mathbf{r} = \left( r_1, \ldots, r_{|P \cup O|} \right)^\mathsf{T}$ for $P \cup O$.

**Step 2** Select the smallest $N$ elements of $\mathbf{r}$ and their corresponding solutions for the new $\boldsymbol{\pi}$ and the new generation of $P$, respectively.

**Step 3** Identify the best solutions for the $N$ subproblems from $P \cup O$ for the new generation of $A$.

Given a decision vector $\mathbf{x}^k \in P \cup O$, we let $R_{i,j}^k$ be the ascending rank of $g_i^{bdn}(\mathbf{x}^k | \mathbf{w}^{i,j}, \mathbf{z}^{r1}, \mathbf{z}^{r2}, \alpha)$ within $\left\{ g_i^{bdn}(\mathbf{x} | \mathbf{w}^{i,j}, \mathbf{z}^{r1}, \mathbf{z}^{r2}, \alpha) \mid \mathbf{x} \in P \cup O \right\}$. $r_k$ is formulated as $r_k = \min_{(i,j) \in I} \{ R_{i,j}^k \}$ where $I = \{ (i,j) | i = 1 \ldots, m, j = 1, \ldots, \iota \}$. $\mathbf{x}^k$ with a smaller $r_k$ implies a higher quality.

## 4 Experimental Studies

### 4.1 Experimental Setup

**Instances.** We proposed 4 scalable test problems denoted as TN1-TN4. The feasible objective regions of TN1 and TN2 are shown in Figure 2a and Figure 2e. The $PF$ of TN1 is a $(m-1)$-dimensional simplex. TN2 has a concave $PF$ constructed by two $(m-1)$-dimensional simplices. TN3 and TN4 are the modified versions of TN1 and TN2 respectively. Their objective functions have different value ranges. Details of TN1-TN4 are available in Appendix B. Moreover, we select problems with different shapes of feasible objective regions, including 6 existing test problems [46, 47, 48, 49] (DTLZ3, mDTLZ3, MaF2, DTLZ5, IMOP4, and IMOP6) and 4 real-world problems [50, 51, 52] (MP-DMP, ML-DMP, RE3-4-7, and RE5-3-1). We consider test problems with 3, 5, and 8 objectives, which accordingly have 8, 12, and 16 variables. The $PF$s of RE3-4-7 and RE5-3-1 are unknown and represented by their current best solution sets.

**Comparison Algorithms[2].** BDNE is compared with 2 representative heuristic algorithms for black-box MOPs: ECR-NSGA-II [54] (based on SF1) and DNPE [34] (based on EP6). ECR-NSGA-II is parameter-free. DNPE is configured according to the corresponding reference (*i.e.*, $\lambda = 100$). For BDNE, we set $\mu = 100$ and $\tau_l = 200$. Then $\tau_u = 14$ according to the setting of $\tau_l$ and $FE_{max}$. The general algorithm settings are summarized in Table 2, where $FE_{max}$ means maximum

Table 2: General algorithm settings.

| $m$ | $N$ | $FE_{max}$ | Operator |
|-----|-----|------------|----------|
| 3 | 60 | 180,000 | SBX+PM |
| 5 | 100 | 300,000 | [53] |
| 8 | 160 | 480,000 | |

number of function evaluations. Each algorithm is executed 30 times on each instance. Additionally, the computer resources and algorithm runtimes can be found in Appendix C.

**Performance Metric.** The estimated nadir objective vector is extracted by Eq. (2) where $S$ is the final population. Let $\hat{z}_i^{nad}$ be the estimated nadir objective vector. The error metric value is computed by $E = \sqrt{\sum_{i=1}^m \left( (z_i^{nad} - \hat{z}_i^{nad}) / (z_i^{nad} - z_i^{ide}) \right)^2}$. The result table records the mean metric value

---

[2]The source code is available at `https://github.com/EricZheng1024/BDNE`.

Table 3: Comparisons of error metric values among ECR-NSGA-II, DNPE, and BDNE.

| Problem | $m$ | ECR-NSGA-II | $\Delta$ | DNPE | $\Delta$ | BDNE |
|---|---|---|---|---|---|---|
| TN1 | 3 | $2.23_{\pm 1.36}(3)$- | -2.23 | $0.203_{\pm 1.58e\text{-}07}(2)$- | -0.203 | $3.14e\text{-}16_{\pm 6.65e\text{-}16}(1)$ |
|  | 5 | $5.28_{\pm 2.11}(3)$- | -5.28 | $0.171_{\pm 1.12e\text{-}06}(2)$- | -0.171 | $0.000273_{\pm 0.000741}(1)$ |
|  | 8 | $12.7_{\pm 1.05}(3)$- | -12.7 | $0.168_{\pm 4.15e\text{-}06}(2)$- | -0.146 | $0.0221_{\pm 0.0331}(1)$ |
| TN2 | 3 | $1.46_{\pm 0.949}(3)$- | -1.46 | $0.223_{\pm 3.54e\text{-}07}(2)$- | -0.223 | $0.000172_{\pm 0.000535}(1)$ |
|  | 5 | $4.77_{\pm 1.91}(3)$- | -4.77 | $0.333_{\pm 4.86e\text{-}10}(2)$- | -0.325 | $0.00803_{\pm 0.0233}(1)$ |
|  | 8 | $12.2_{\pm 0.843}(3)$- | -12.2 | $0.333_{\pm 3.96e\text{-}10}(2)$- | -0.315 | $0.0185_{\pm 0.0204}(1)$ |
| TN3 | 3 | $2.15_{\pm 1.43}(3)$- | -2.15 | $0.263_{\pm 4.02e\text{-}08}(2)$- | -0.263 | $2.39e\text{-}16_{\pm 2.32e\text{-}16}(1)$ |
|  | 5 | $4.7_{\pm 1.77}(3)$- | -4.69 | $0.263_{\pm 1.85e\text{-}08}(2)$- | -0.262 | $0.000669_{\pm 0.00203}(1)$ |
|  | 8 | $12.7_{\pm 0.718}(3)$- | -12.7 | $0.263_{\pm 1.2e\text{-}08}(2)$- | -0.243 | $0.0195_{\pm 0.0263}(1)$ |
| TN4 | 3 | $1.35_{\pm 0.88}(3)$- | -1.35 | $0.337_{\pm 4.04e\text{-}08}(2)$- | -0.337 | $4.92e\text{-}05_{\pm 5.59e\text{-}05}(1)$ |
|  | 5 | $5.34_{\pm 2.17}(3)$- | -5.34 | $0.337_{\pm 1.9e\text{-}08}(2)$- | -0.335 | $0.00208_{\pm 0.00422}(1)$ |
|  | 8 | $12.3_{\pm 0.951}(3)$- | -12.3 | $0.337_{\pm 1.42e\text{-}08}(2)$- | -0.31 | $0.0268_{\pm 0.031}(1)$ |
| DTLZ2 | 3 | $0.00268_{\pm 0.00277}(3)$- | -0.00268 | $4.29e\text{-}10_{\pm 3.33e\text{-}10}(2)$- | -2.48e-10 | $1.81e\text{-}10_{\pm 2.82e\text{-}10}(1)$ |
|  | 5 | $0.0269_{\pm 0.0171}(3)$- | -0.0269 | $6.04e\text{-}10_{\pm 5.2e\text{-}10}(2)$- | -4.36e-10 | $1.68e\text{-}10_{\pm 1.86e\text{-}10}(1)$ |
|  | 8 | $0.233_{\pm 0.167}(3)$- | -0.233 | $5.91e\text{-}10_{\pm 3.86e\text{-}10}(2)$- | -2.04e-10 | $3.87e\text{-}10_{\pm 5.31e\text{-}10}(1)$ |
| mDTLZ2 | 3 | $0.171_{\pm 0.0995}(3)$- | -0.164 | $0.0209_{\pm 6.14e\text{-}06}(2)$- | -0.0143 | $0.00658_{\pm 0.00105}(1)$ |
|  | 5 | $0.581_{\pm 0.116}(3)$- | -0.576 | $0.0112_{\pm 9.19e\text{-}06}(2)$- | -0.00648 | $0.0047_{\pm 0.000487}(1)$ |
|  | 8 | $0.826_{\pm 0.00972}(3)$- | -0.823 | $0.00859_{\pm 1.25e\text{-}05}(2)$- | -0.005 | $0.00358_{\pm 0.00499}(1)$ |
| MaF2 | 3 | $0.0243_{\pm 0.018}(3)$- | -0.0154 | $0.00888_{\pm 2.9e\text{-}11}(2)$- | -2.1e-06 | $0.00888_{\pm 1.15e\text{-}05}(1)$ |
| DTLZ5 | 3 | $0.005_{\pm 0.00704}(3)$- | -0.005 | $5.81e\text{-}10_{\pm 4.6e\text{-}10}(2)$- | -5.81e-10 | $1.43e\text{-}13_{\pm 6.6e\text{-}13}(1)$ |
| IMOP4 | 3 | $7.69e\text{-}05_{\pm 0.00023}(2)$- | -7.69e-05 | $0.748_{\pm 0.0832}(3)$- | -0.748 | $4.16e\text{-}17_{\pm 1.09e\text{-}16}(1)$ |
| IMOP6 | 3 | $0.0139_{\pm 0.0115}(3)$- | -0.0139 | $0.00125_{\pm 2.31e\text{-}08}(2)$- | -0.00125 | $3.2e\text{-}15_{\pm 8.49e\text{-}15}(1)$ |
| MP-DMP | 3 | $0.369_{\pm 0.168}(3)$- | -0.361 | $0.23_{\pm 8.03e\text{-}06}(2)$- | -0.222 | $0.00777_{\pm 2.85e\text{-}05}(1)$ |
|  | 5 | $0.357_{\pm 0.0871}(2)$- | -0.344 | $0.767_{\pm 2.17e\text{-}06}(3)$- | -0.754 | $0.0127_{\pm 4.88e\text{-}05}(1)$ |
| ML-DMP | 3 | $80.8_{\pm 0.0603}(3)$- | -80.7 | $0.00699_{\pm 2.82e\text{-}05}(1)$+ | 0.00917 | $0.0162_{\pm 8.56e\text{-}05}(2)$ |
|  | 5 | $85.1_{\pm 7.01}(3)$- | -85.1 | $0.561_{\pm 5.76e\text{-}06}(2)$- | -0.521 | $0.0397_{\pm 0.072}(1)$ |
| RE3-4-7 | 3 | $0.133_{\pm 0.00299}(2)$- | -0.0538 | $0.545_{\pm 2.61e\text{-}06}(3)$- | -0.466 | $0.0787_{\pm 0.00862}(1)$ |
| CRE5-3-1 | 5 | $24.8_{\pm 61.9}(3)$= | -24.6 | $0.191_{\pm 7.18e\text{-}08}(1)$+ | 8.58e-08 | $0.191_{\pm 9.99e\text{-}09}(2)^{\dagger}$ |
| Total +/=/- |  | 0/1/27 |  | 2/0/26 |  | \ |
| Average rank |  | 2.8929(3) |  | 2.0357(2) |  | 1.0714(1) |

$^{\dagger}$ The approximate $PF$ may be the reason for the large error on CRE5-3-1 since the results of DNPE and BDNE are similar.

and the standard deviation across all runs for each instance. The performance rank on each instance is inside parentheses. "+", "=" or "-" denotes that the performance of the corresponding algorithm is statistically better than, similar to, or worse than that of BDNE based on Wilcoxon's rank sum test at 0.05 significant level. "$\Delta$" indicates the gap between the mean metric value of the corresponding algorithm and that of BDNE. The best mean metric values are also emphasized.

## 4.2 Results

Table 3 shows the statistical results on 28 instances. Across most instances, BDNE outperforms the other algorithms and has mean metric values below $5\%$. In Figure 2, we can see that BDNE accurately approximates at least one critical point for each objective, consistent with our theoretical analysis. DNPE ranks second. DNPE has competitive results on MaF2, DTLZ5, DTLZ2, mDTLZ2, IMOP6, 3-objective ML-DMP, and CRE5-3-1. We can infer that the extreme points determined by DNPE are very close to the critical points of these MOPs. Nevertheless, the performance of DNPE deteriorates on other instances. Besides, comparing the results between TN1 and TN3, value ranges of objective functions significantly impact DNPE but hardly affect BDNE. ECR-NSGA-II achieves the worst overall performance. Moreover, it has large standard deviations on many instances, which indicates its highly unstable performance. This is because ECR-NSGA-II suffers from dominance-resistant

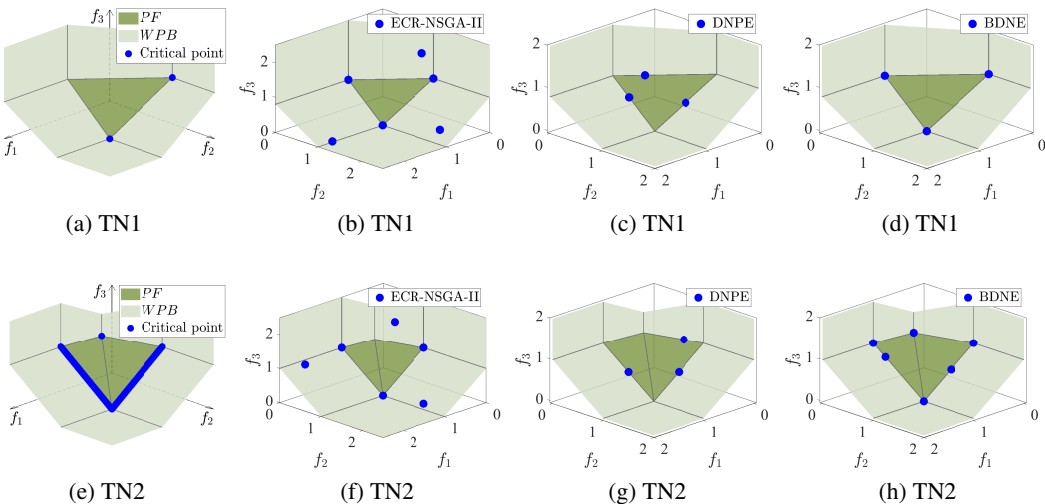

Figure 2: Plots of the final solution sets with median error metric values.

solutions. For example, ECR-NSGA-II retains solutions close to the $WPB$ in Figure 2b and Figure 2f. Each of these solutions has at least one inferior objective function value.

In general, the two algorithms have huge performance gaps compared with BDNE. The effectiveness and superiority of BDNE are demonstrated. More experiments are presented in Appendix E.

## 5  Conclusion, Limitation, and Future Work

**Conclusion.** In this paper, we have revealed the deficiency of existing methods in estimating the nadir objective vector. Specifically, exact methods suffer from limited applicability and high computational costs, while the irregular $PF$ and the $WPB$ can cause significant challenges for heuristic methods. We have proposed a new scalarization method, which can define specific boundary subproblems to find the nadir objective vector under mild conditions. We have formulated $m$ BLOPs using boundary subproblems and designed a corresponding algorithm framework called BDNE. We have also conducted experimental studies to validate the effectiveness of BDNE. In experiments, BDNE adopts evolutionary algorithms and effectively approximates the nadir objective vectors of various black-box MOPs.

**Limitation and Future Work.** The estimated nadir objective vector can be obtained beforehand or improved with the optimization process. In the paper, we present BDNE as an independent algorithm (*i.e.*, estimate the nadir objective vector beforehand). In the future, we will investigate how to integrate BDNE into the iteration of an algorithm to enhance its overall performance (see Appendix E.4 for some pilot studies). Furthermore, BDNE has been implemented in a general manner. We plan to refine BDNE for specific applications, including multi-objective discrete optimization problems. Potential societal impacts can be found in Appendix D.

## Acknowledgments and Disclosure of Funding

This work was supported by the National Natural Science Foundation of China (Grant No. 62106096 and Grant No. 62476118), the Natural Science Foundation of Guangdong Province (Grant No. 2024A1515011759), the National Natural Science Foundation of Shenzhen (Grant No.JCYJ20220530113013031).

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

This is an appendix for "Boundary Decomposition for Nadir Objective Vector Estimation". Specifically, we provide:

- Detailed analyses of heuristic methods (Appendix A);
- Mathematical models of the proposed benchmark problems, *i.e.*, TN1-TN4 (Appendix B);
- The computational environment, algorithm runtimes, and the complexity analysis of BDNE (Appendix C);
- The potential societal impact of this work (Appendix D);
- More experimental results and analyses including ablation studies, the impact of $\mu$, and the comparison with other heuristic methods (Appendix E);
- Properties of the cone domination and proofs of theorems and corollaries (Appendix F).

## A    Deficiencies of Heuristic Methods

Existing heuristic methods for the nadir objective vector estimation can be divided into two categories [15]: 1) straightforward methods and 2) extreme-point-based methods. We use an example $PF$ to illustrate the drawbacks of heuristic methods in estimating the nadir objective vector. As shown in Figure 3, the $PF$ consists of two triangles and is concave. We uniformly sample 225 points along the $PF$ in order to include sufficient critical points. We use these sampled points as each method's population and test its performance in estimating the nadir objective vector.

The experimental results are summarized in Table 4 and plotted in Figure 4. We can observe from Table 4 that all methods incorrectly estimate the nadir objective vector. Specifically, Figure 4 show that all methods except for SF4 acquire the critical points of $f_1$ and $f_2$ but miss that of $f_3$. As shown in Figure 3, the critical point of $f_3$ is distant from every axis and thereby difficult to obtain by these methods.

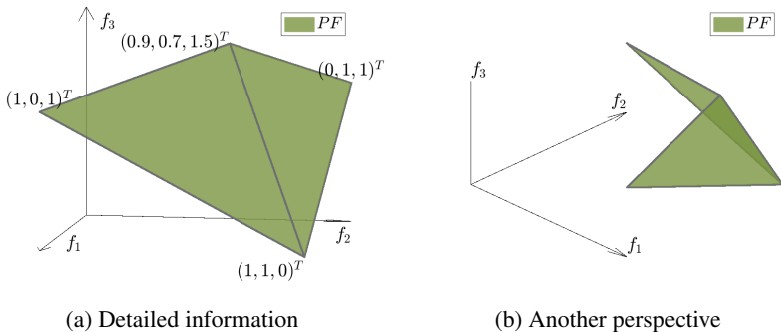

(a) Detailed information                     (b) Another perspective

Figure 3: Illustrations of an example $PF$. The critical points of $f_1$ are convex combinations of $(1, 0, 1)^\intercal$ and $(1, 1, 0)^\intercal$. The critical points of $f_2$ are convex combinations of $(0, 1, 1)^\intercal$ and $(1, 1, 0)^\intercal$. The critical point of $f_3$ is $(0.9, 0.7, 1.5)^\intercal$. The nadir objective vector is $(1, 1, 1.5)^\intercal$.

Table 4: Estimated nadir objective vectors of existing heuristic methods. The nadir objective vector is $(1, 1, 1.5)^\intercal$. "SFx" denotes the straightforward method; "EPx" denotes the extreme-point-based method. Parameters settings: the minimum number of selected points is set to 10 for SF3 and SF4; $\lambda$ in EP6 is set to 100.

| Method | Result |
|---|---|
| SF2 [24] | $(1, 1, 1.14)^\intercal$ |
| SF3 [25] | $(1, 1, 1)^\intercal$ |
| SF4 [26] | $(0.88, 0.96, 1.2)^\intercal$ |
| EP1 [29] | $(1, 1, 1)^\intercal$ |
| EP2 [28] | $(1, 1, 1)^\intercal$ |
| EP3 ($L_2$) [31] | $(1, 1, 1.18)^\intercal$ |
| ($L_\infty$) [32] | $(1, 1, 1.39)^\intercal$ |
| EP4 [33] | $(1, 1, 1.18)^\intercal$ |
| EP5 [9] | $(1, 1, 1.39)^\intercal$ |
| EP6 [34] | $(1, 1, 1.18)^\intercal$ |

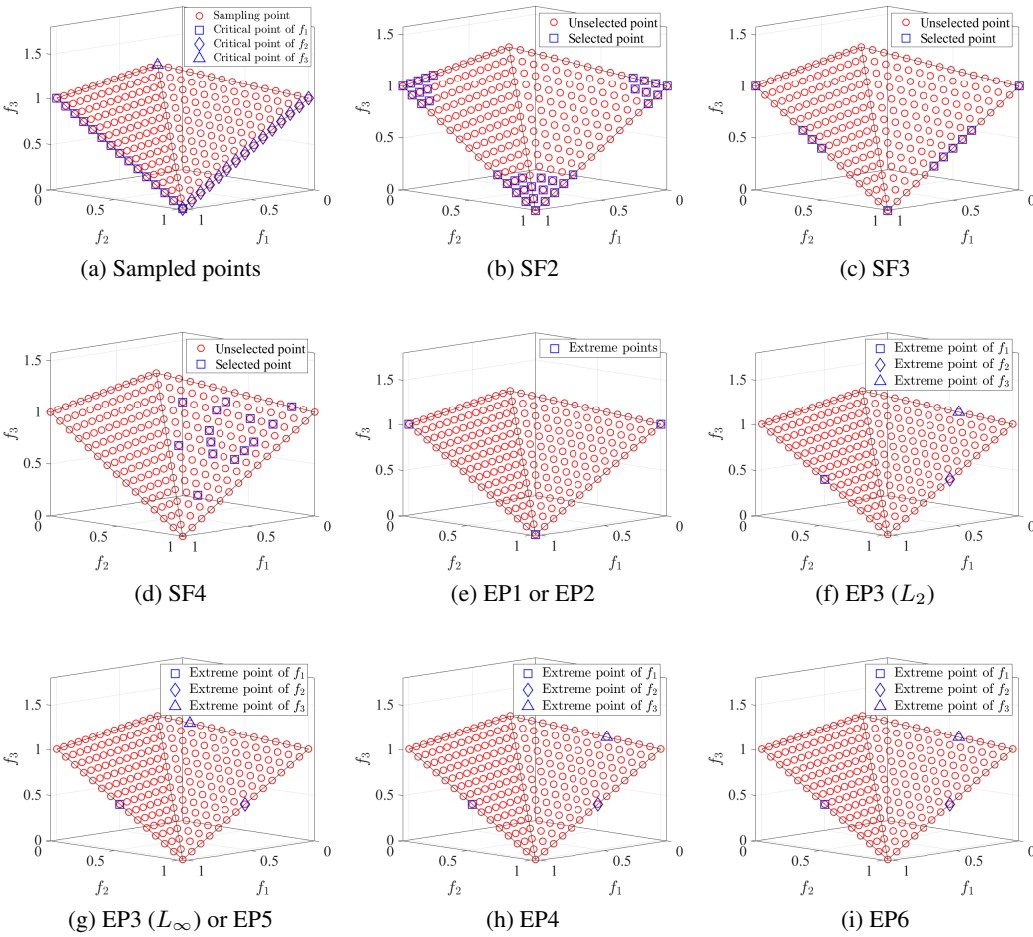

Figure 4: Illustrations of selected points obtained by existing heuristic methods.

# B Test Problem with Adjustable Critical Points

## B.1 Test Problem Generator

The existing test problems [55] can not have various critical points. Many test problems share similar (or even identical) nadir objective vector configurations. For example, most problems of DTLZ [46] and WFG [56] are constructed from the unit simplex (*i.e.*, equilateral triangle). The critical points of the unit simplex are its vertices. We design two problem generators that have controllable critical points.

Each objective function of the proposed test problem generators takes the following form

$$f_i(\mathbf{x}) = s_i h_i(\mathbf{x}_I)(1 + g_i(\mathbf{x}_{II})), \tag{13}$$

where $s_i$ is a constant such that different objectives can have different ranges of values; $\mathbf{x} \in \Omega = [0,1]^n \subset \mathbb{R}^n$, and $\mathbf{x}_I = (x_1, \ldots, x_m)^\mathsf{T}$ and $\mathbf{x}_{II} = (x_{m+1}, \ldots, x_n)^\mathsf{T}$ are subvectors of $\mathbf{x}$. $g_i(\mathbf{x}_{II}) \geq 0$ is called the distance function, which specifies as

$$g_i(\mathbf{x}_{II}) = 10 \cdot \sum_{j \in J_i} (x_j - 0.3)^2,$$

$$J_i = \left\{ m+i, 2m+i, \ldots, \left\lfloor \frac{n-i}{m} \right\rfloor m + i \right\}. \tag{14}$$

$h_i(\mathbf{x}_I)$ is termed as the position function, which determines the shape of the $PF$.

### B.1.1 Generator 1

The $PF$ is a simplex. Firstly, we define $\mathbf{y}(\mathbf{x}_I) : [0,1]^m \to [0,1]^m$ as a base. It returns an $m$-dimensional vector $(y_1, \ldots, y_m)^\mathsf{T}$ as follows

$$y_i = \frac{x_i}{\sum\limits_{j=1}^{m} x_j} \text{ for } i = 1, \ldots, m. \tag{15}$$

$\{\mathbf{y}(\mathbf{x}_I) | \mathbf{x}_I \in [0,1]^m\}$ is an $(m-1)$-dimensional unit simplex. Then the position function value vector is calculated by

$$\mathbf{h}(\mathbf{x}_I) = V\mathbf{y}(\mathbf{x}_I), \tag{16}$$

where $V = [\mathbf{v}^1, \ldots, \mathbf{v}^m]$ determines vertices of the $PF$. When $\mathbf{s} = (1, \ldots, 1)^\mathsf{T}$, $\mathbf{v}^i$ is a vertex of the $PF$. Any $\mathbf{v}^i$ should be non-dominated. $V$ is the particular parameter to be specified. The critical points depend on the settings of $V$.

### B.1.2 Generator 2

The $PF$ is two adjacent simplices. We modify the base of Generator 1 to

$$y_i = \begin{cases} \frac{x_i}{2x_m + \sum\limits_{j=1}^{m-1} x_j}, & i = 1, \ldots, m-1 \wedge x_m \leq 0.5, \\[3ex] \frac{x_i}{2(1-x_m) + \sum\limits_{j=1}^{m-1} x_j}, & i = 1, \ldots, m-1 \wedge x_m > 0.5, \\[3ex] \frac{2x_i}{2x_m + \sum\limits_{j=1}^{m-1} x_j}, & i = m \wedge x_m \leq 0.5, \\[3ex] \frac{2(1-x_i)}{2(1-x_m) + \sum\limits_{j=1}^{m-1} x_j}, & i = m \wedge x_m > 0.5. \end{cases} \tag{17}$$

Letting

$$\begin{aligned} u_j > 1, j = 3, \\ u_j < 1, j \neq 3, \end{aligned} \tag{18}$$

and

$$v_j^i = \begin{cases} 1, & j \neq i, \\ 0, & j = i, \end{cases} \tag{19}$$

the position function value vector is

$$\mathbf{h}(\mathbf{x}_I) = \begin{cases} V\mathbf{y}(\mathbf{x}_I), & x_m \leq 0.5, \\ V'\mathbf{y}(\mathbf{x}_I), & x_m > 0.5, \end{cases} \tag{20}$$

where

$$\begin{aligned} V &= [\mathbf{v}^1, \mathbf{u}, \mathbf{v}^5, \dots, \mathbf{v}^m, \mathbf{v}^2], \\ V' &= [\mathbf{v}^1, \mathbf{u}, \mathbf{v}^5, \dots, \mathbf{v}^m, \mathbf{v}^4]. \end{aligned} \tag{21}$$

The $PF$ of this generator consists of two simplices. Assume that $\mathbf{s} = (1, \dots, 1)^\intercal$. The vertices of two simplices are the column vectors of $V$ and $V'$ respectively. The difference between $V$ and $V'$ is the last column. That is to say, two simplices share many common edges and they are adjacent. $\mathbf{u}$ is the particular parameter to be specified. The unique critical point of $f_3$ is $\mathbf{u}$. The critical points of other objectives are the same as the inverted unit simplex whose objective vectors on any edge are critical points.

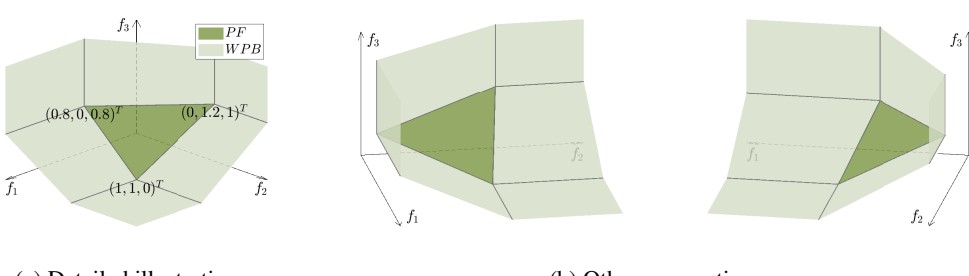

(a) Detailed illustration        (b) Other perspectives

Figure 5: The feasible objective region of 3-objective TN1.

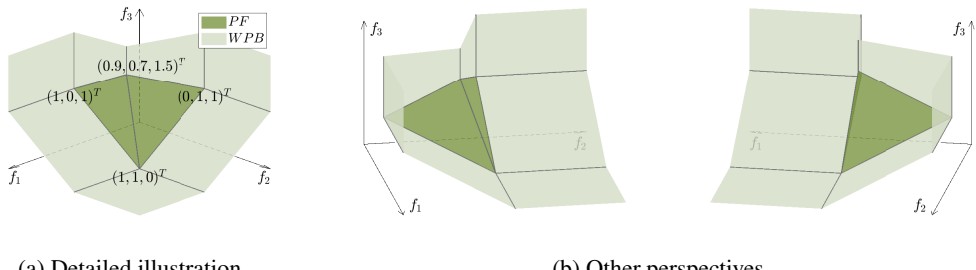

(a) Detailed illustration        (b) Other perspectives

Figure 6: The feasible objective region of 3-objective TN2.

## B.2 Proposed Test Problems

We use the proposed test problem generators to obtain 4 scalable MOPs denoted as TN1-TN4. We first specify 1 matrix and 3 vectors:

- 
$$(V_1)_{ij} = \begin{cases} 0, & \text{if } i = j, \\ 1.2, & \text{if } i = 2 \wedge j = 1, \\ 0.8, & \text{if } (i = 1 \vee i = 3) \wedge j = 2, \\ 1, & \text{otherwise}; \end{cases} \tag{22}$$

- $\mathbf{s}^1 = (1, \dots, 1)^\intercal$;
- if $m = 3$, $\mathbf{s}^2 = (1, 100, 10000)^\intercal$; otherwise, $\mathbf{s}^2 = (1, 100, 10000, 1, \dots, 1)^\intercal$;
- if $m = 3$, $\mathbf{u}^1 = (0.9, 0.7, 1.5)^\intercal$; otherwise, $\mathbf{u}^1 = (0.9, 0.7, 1.5, 0.9, \dots, 0.9)^\intercal$.

Then the test problems are summarized in Table 5. We can find that $V_1$ is obtained by modifying such matrix: an $m \times m$ square matrix with zeros on the main diagonal and ones elsewhere. This matrix implies that the $PF$ is an inverted unit simplex whose objective vectors on any edge are critical points (*e.g.*, mDTLZ1 [48]). The modified vertices are the first and the second ones and the modification performs across $f_1$ to $f_3$. Consequently, the critical points of $f_1$, $f_2$, and $f_3$ are some vertices only. For example, we show the feasible objective region of the 3-objective TN1 in Figure 5. Vertices $(0, 1.2, 1)^\intercal$ and $(1, 1, 0)^\intercal$ are the only critical points. TN2 is the specification of the studied case in Section 2. More specifically, as illustrated in Figure 6, we can find that the critical point of $f_3$ is unsupported (*i.e.*, there exists convex combinations of objective vectors that dominate the critical point [57]), making the $PF$ exhibit a concave shape. Besides, the critical point of $f_3$ is distant from all axes and not optimal for any objective. TN3 and TN4 are extended from TN1 and TN2 respectively. They have huge magnitude differences between some objective functions.

Table 5: Parameter settings of TN1-TN4.

| Problem | Generator | Parameters | Nadir objective vector |
|---------|-----------|------------|------------------------|
| TN1 | 1 | $\mathbf{s} = \mathbf{s}^1, V = V_1$ | $(1, 1.2, 1, \ldots, 1)^\intercal$ |
| TN3 | 2 | $\mathbf{s} = \mathbf{s}^1, \mathbf{u} = \mathbf{u}^1$ | $(1, 1, 1.5, 1, \ldots, 1)^\intercal$ |
| TN3 | 1 | $\mathbf{s} = \mathbf{s}^2, V = V_1$ | $(1, 120, 10000, 1, \ldots, 1)^\intercal$ |
| TN4 | 2 | $\mathbf{s} = \mathbf{s}^2, \mathbf{u} = \mathbf{u}^1$ | $(1, 100, 15000, 1, \ldots, 1)^\intercal$ |

Figure 5 and Figure 6 also indicate these 4 problems process the $WPB$. The $WPB$ has evident dominance resistance [23]. That is, the objective vectors of the $WPB$ are usually non-dominated in the population, especially in many objective cases. The reason is that each of them can only improve $k$ ($1 \leq k \leq m - 1$) objectives to obtain an objective vector dominating it. Thus, they are difficult to distinguish from the population. The $WPB$ causes the population to suffer poor convergence and diversity such that the nadir objective vector is hard to estimate. As found in [48], the objective vectors with $k = 1$ on the $WPB$ exhibit severe dominance resistance, even when presenting in 3-objective cases. Unfortunately, how to cope with the $WPB$ is still an open question [58].

## C   Runtime

In experiments, our codes rely on the MATLAB-based platform called PlatEMO [59], which is freely available for research purposes (see `https://github.com/BIMK/PlatEMO`). Experiments are executed on a computer equipped with two 3.00-GHz Intel Xeon Gold 6248R CPUs (48 cores in total), an NVIDIA T400 GPU, and 128GB of RAM. We report the runtime for each algorithm on each instance in Table 6. DNPE exhibits the shortest runtime, followed by ECR-NSGA-II. BDNE has a longer runtime than these two representative algorithms.

We want to emphasize the runtime for BDNE strongly depends on the implementation (*e.g.*, the chosen solvers). We adopt an effective selection procedure stated in [60], which can facilitate the application of BDNE to various problems. The main loop of our BDNE implementation has a time complexity of $O\left(N^2 \log(N)\right)$ where $N$ is the population size. The specific analysis is provided as follows. The reproduction procedure has a time complexity of $O(N)$. The calculation of $\mathbf{r}$ governs the complexity of the selection procedure, which has $O\left(N^2 \log(N)\right)$. The adjustment of boundary weight vectors with $O\left(N \log\left(\frac{N}{m}\right)\right)$ is executed periodically. Therefore, The selection procedure governs the overall complexity, and the main loop has the complexity $O\left(N^2 \log(N)\right)$. We can significantly reduce the complexity by using a simple selection procedure or choosing an alternative solver, which deserves further investigation.

Table 6: Comparisons of runtimes among ECR-NSGA-II, DNPE, and BDNE.

| Problem | $m$ | ECR-NSGA-II | $\Delta$ | DNPE | $\Delta$ | BDNE |
|---|---|---|---|---|---|---|
| TN1 | 3 | $10.6_{\pm0.335}(2)+$ | 11 | $7.17_{\pm0.46}(1)+$ | 14.5 | $21.6_{\pm0.401}(3)$ |
| | 5 | $15.6_{\pm0.512}(2)+$ | 23.4 | $11.1_{\pm0.515}(1)+$ | 27.9 | $39_{\pm0.613}(3)$ |
| | 8 | $31.2_{\pm0.641}(2)+$ | 45.5 | $21.1_{\pm0.876}(1)+$ | 55.6 | $76.7_{\pm0.998}(3)$ |
| TN2 | 3 | $9.5_{\pm0.259}(2)+$ | 12 | $7.26_{\pm0.342}(1)+$ | 14.2 | $21.5_{\pm0.415}(3)$ |
| | 5 | $16.4_{\pm0.44}(2)+$ | 23.4 | $11.5_{\pm0.366}(1)+$ | 28.4 | $39.9_{\pm0.552}(3)$ |
| | 8 | $32.1_{\pm0.622}(2)+$ | 45.9 | $21.7_{\pm0.67}(1)+$ | 56.3 | $77.9_{\pm0.836}(3)$ |
| TN3 | 3 | $11.8_{\pm0.795}(2)+$ | 14.2 | $6.31_{\pm0.261}(1)+$ | 19.7 | $26_{\pm0.641}(3)$ |
| | 5 | $17.8_{\pm1.3}(2)+$ | 26.2 | $10.8_{\pm0.383}(1)+$ | 33.2 | $43.9_{\pm1.02}(3)$ |
| | 8 | $35.1_{\pm2.09}(2)+$ | 49.7 | $20.3_{\pm0.531}(1)+$ | 64.4 | $84.7_{\pm1.77}(3)$ |
| TN4 | 3 | $11_{\pm0.748}(2)+$ | 13.2 | $7.02_{\pm0.344}(1)+$ | 17.2 | $24.2_{\pm0.564}(3)$ |
| | 5 | $18.6_{\pm1.23}(2)+$ | 25.5 | $11.5_{\pm0.408}(1)+$ | 32.7 | $44.1_{\pm0.733}(3)$ |
| | 8 | $35.9_{\pm2.28}(2)+$ | 35 | $21.2_{\pm0.678}(1)+$ | 49.7 | $70.9_{\pm0.738}(3)$ |
| DTLZ2 | 3 | $7.29_{\pm0.171}(2)+$ | 12.8 | $4.05_{\pm0.235}(1)+$ | 16.1 | $20.1_{\pm0.474}(3)$ |
| | 5 | $10.8_{\pm0.168}(2)+$ | 25.8 | $5.43_{\pm0.23}(1)+$ | 31.2 | $36.7_{\pm0.551}(3)$ |
| | 8 | $16.9_{\pm0.318}(2)+$ | 53.4 | $8.62_{\pm0.3}(1)+$ | 61.7 | $70.3_{\pm1}(3)$ |
| mDTLZ2 | 3 | $9.8_{\pm0.204}(2)+$ | 13.2 | $6.26_{\pm0.259}(1)+$ | 16.7 | $23_{\pm0.476}(3)$ |
| | 5 | $17.9_{\pm0.286}(2)+$ | 25.2 | $11_{\pm0.409}(1)+$ | 32.1 | $43.1_{\pm0.735}(3)$ |
| | 8 | $35_{\pm0.482}(2)+$ | 49.6 | $21.1_{\pm0.712}(1)+$ | 63.4 | $84.5_{\pm1.14}(3)$ |
| MaF2 | 3 | $9.22_{\pm0.493}(2)+$ | 12.6 | $4.35_{\pm0.193}(1)+$ | 17.5 | $21.8_{\pm0.453}(3)$ |
| DTLZ5 | 3 | $7.75_{\pm0.29}(2)+$ | 13.6 | $4.21_{\pm0.269}(1)+$ | 17.1 | $21.3_{\pm0.354}(3)$ |
| IMOP4 | 3 | $7.73_{\pm0.293}(2)+$ | 13.5 | $4.18_{\pm0.218}(1)+$ | 17.1 | $21.3_{\pm1.66}(3)$ |
| IMOP6 | 3 | $7.88_{\pm0.315}(2)+$ | 13.8 | $4.22_{\pm0.205}(1)+$ | 17.4 | $21.7_{\pm1.95}(3)$ |
| MP-DMP | 3 | $7.29_{\pm0.173}(2)+$ | 14.2 | $4.31_{\pm0.233}(1)+$ | 17.2 | $21.5_{\pm1.87}(3)$ |
| | 5 | $10.7_{\pm0.203}(2)+$ | 25.9 | $5.01_{\pm0.258}(1)+$ | 31.6 | $36.7_{\pm2.96}(3)$ |
| ML-DMP | 3 | $7.3_{\pm0.191}(2)+$ | 14.7 | $4.54_{\pm0.287}(1)+$ | 17.5 | $22_{\pm1.95}(3)$ |
| | 5 | $12.2_{\pm0.493}(2)+$ | 28.3 | $7.75_{\pm0.449}(1)+$ | 32.8 | $40.5_{\pm3.28}(3)$ |
| RE3-4-7 | 3 | $11.4_{\pm0.412}(2)+$ | 13.6 | $7.28_{\pm0.326}(1)+$ | 17.7 | $25_{\pm2.14}(3)$ |
| CRE5-3-1 | 5 | $17.2_{\pm1.59}(2)+$ | 30.7 | $14.7_{\pm0.64}(1)+$ | 33.2 | $47.9_{\pm4.01}(3)$ |
| Total +/=/- | | 28/0/0 | | 28/0/0 | | \ |
| Average rank | | 2(2) | | 1(1) | | 3(3) |

# D   Potential societal impact

Many real-world problems, such as logistics dispatch, printed-circuit board assembly, efficient chemical reaction, and Internet of Things service, require simultaneous optimization of multiple conflicting objectives. The nadir objective vector mainly has two strengths in multi-objective optimization: 1) it can normalize the objective space together with the ideal objective vector and 2) it provides a range for decision-makers or algorithms to more easily explore different trade-offs. Our proposed BDNE is good for better nadir objective vector estimation in terms of theoretical guarantee and wide applicability. BDNE can be applied to various problems by employing appropriate solvers. Furthermore, the parameter $\mu$, which indicates the tolerance of decision-makers for trade-offs, could also be an important component in developing a more user-friendly multi-criteria decision system.

On the downside, relying solely on the nadir objective vector for decision-making can be risky. Decision-makers should use additional information for a more informed and comprehensive decision. Moreover, the leakage of the nadir objective vector might inadvertently expose problem information and user preferences. This is particularly troublesome in certain applications where the confidentiality of such information is paramount, indicating a need for caution to avoid these situations. In addition, there is no guarantee that the evolutionary algorithm version can approximate the nadir objective vector within a desired timeframe. This might delay decision-making processes or lead to decisions based on incomplete or inadequate information.

# E  More Experiments

We conduct a more comprehensive empirical study in this section. We omit comparisons with existing exact methods, as they are implemented exclusively on the discrete MOP (where the exact solver is available) and are nearly impossible to apply to other types of MOPs. Nevertheless, we will propose the implementation of BDNE for the discrete MOP. Audiences can follow our work at https://github.com/EricZheng1024/BDNE.

## E.1  Ablation Studies

We investigate 3 schemes in BDNE. "CFF" denotes the coping strategy for flat fitness (see Section 3.2). "TBW" denotes the transformation of the best boundary weight vector (see Step 2 of the upper-level optimization in Section 3.3). "BSU" denotes the boundary subproblems using unit vectors as their weight vectors (see the upper-level optimization in Section 3.3).

Table 7 presents the experimental results. Removing CFF or TBW merely does not evidently affect the performance of BDNE. However, the performance deteriorates when both schemes are absent. Particularly, BDNE without CFF and TBW is less effective on MOPs with complicated feasible regions, *i.e.*, TN2 and TN4. BDNE outperforms BDNE without BSU across most instances, indicating that boundary subproblems using unit vectors are important. Overall, the effectiveness of the two strategies is validated.

Table 7: Comparisons of error metric values among variants of BDNE.

| Problem | $m$ | w/o CFF | w/o TBW | w/o CFF & TBW | w/o BSU | BDNE |
|---|---|---|---|---|---|---|
| TN1 | 3 | $2.5\text{e-}16_{\pm 2.92\text{e-}16}(2)=$ | $1.8\text{e-}16_{\pm 1.77\text{e-}16}(1)=$ | $2.55\text{e-}16_{\pm 2.17\text{e-}16}(3)=$ | $0.000119_{\pm 0.00065}(5)\text{-}$ | $3.14\text{e-}16_{\pm 6.65\text{e-}16}(4)$ |
| | 5 | $5.5\text{e-}05_{\pm 0.000113}(1)=$ | $0.00277_{\pm 0.00965}(4)=$ | $0.00164_{\pm 0.00528}(3)=$ | $0.0132_{\pm 0.0374}(5)\text{-}$ | $0.000273_{\pm 0.000741}(2)$ |
| | 8 | $0.0208_{\pm 0.0557}(3)=$ | $0.0192_{\pm 0.0369}(2)=$ | $0.0141_{\pm 0.0184}(1)=$ | $0.0398_{\pm 0.0403}(5)\text{-}$ | $0.0221_{\pm 0.0331}(4)$ |
| TN2 | 3 | $0.00016_{\pm 0.000351}(1)=$ | $0.000783_{\pm 0.00195}(4)=$ | $0.000494_{\pm 0.000865}(3)\text{-}$ | $0.00548_{\pm 0.0137}(5)=$ | $0.000172_{\pm 0.000535}(2)$ |
| | 5 | $0.0106_{\pm 0.0375}(5)=$ | $0.00624_{\pm 0.00912}(1)=$ | $0.0103_{\pm 0.0157}(4)=$ | $0.00776_{\pm 0.0124}(2)+$ | $0.00803_{\pm 0.0233}(3)$ |
| | 8 | $0.0351_{\pm 0.0377}(5)=$ | $0.0308_{\pm 0.0283}(4)=$ | $0.0259_{\pm 0.0284}(2)=$ | $0.0287_{\pm 0.0406}(3)=$ | $0.0185_{\pm 0.0204}(1)$ |
| TN3 | 3 | $3.28\text{e-}16_{\pm 3.52\text{e-}16}(4)=$ | $2.37\text{e-}16_{\pm 1.77\text{e-}16}(1)=$ | $2.97\text{e-}16_{\pm 4.53\text{e-}16}(3)=$ | $0.00217_{\pm 0.00995}(5)\text{-}$ | $2.39\text{e-}16_{\pm 2.32\text{e-}16}(2)$ |
| | 5 | $0.000616_{\pm 0.00235}(2)=$ | $0.000568_{\pm 0.00271}(1)=$ | $0.000735_{\pm 0.00228}(4)=$ | $0.0108_{\pm 0.0178}(5)\text{-}$ | $0.000669_{\pm 0.00203}(3)$ |
| | 8 | $0.0192_{\pm 0.0401}(3)=$ | $0.0175_{\pm 0.0211}(2)=$ | $0.0168_{\pm 0.0375}(1)=$ | $0.0381_{\pm 0.0186}(5)\text{-}$ | $0.0195_{\pm 0.0263}(4)$ |
| TN4 | 3 | $0.00011_{\pm 0.000245}(2)=$ | $0.000464_{\pm 0.00106}(4)=$ | $0.00017_{\pm 0.000181}(3)\text{-}$ | $0.00796_{\pm 0.0225}(5)\text{-}$ | $4.92\text{e-}05_{\pm 5.59\text{e-}05}(1)$ |
| | 5 | $0.00288_{\pm 0.0054}(2)=$ | $0.00724_{\pm 0.0134}(4)\text{-}$ | $0.00696_{\pm 0.0119}(3)\text{-}$ | $0.0158_{\pm 0.0416}(5)\text{-}$ | $0.00208_{\pm 0.00422}(1)$ |
| | 8 | $0.0253_{\pm 0.0335}(3)=$ | $0.0227_{\pm 0.0248}(1)=$ | $0.047_{\pm 0.0667}(5)\text{-}$ | $0.0237_{\pm 0.023}(2)=$ | $0.0268_{\pm 0.031}(4)$ |
| DTLZ2 | 3 | $1.07\text{e-}10_{\pm 1.23\text{e-}10}(3)=$ | $2.49\text{e-}10_{\pm 2.36\text{e-}10}(5)=$ | $8.82\text{e-}11_{\pm 9.94\text{e-}11}(2)+$ | $2.2\text{e-}12_{\pm 5.39\text{e-}13}(1)+$ | $1.81\text{e-}10_{\pm 2.82\text{e-}10}(4)$ |
| | 5 | $1.49\text{e-}10_{\pm 1.94\text{e-}10}(3)=$ | $1.03\text{e-}10_{\pm 1.84\text{e-}10}(1)=$ | $1.26\text{e-}10_{\pm 1.4\text{e-}10}(2)=$ | $0.000572_{\pm 0.00166}(5)\text{-}$ | $1.68\text{e-}10_{\pm 1.86\text{e-}10}(4)$ |
| | 8 | $5.38\text{e-}10_{\pm 5.58\text{e-}10}(3)=$ | $4.39\text{e-}10_{\pm 3.99\text{e-}10}(2)=$ | $0.0667_{\pm 0.254}(5)=$ | $0.000282_{\pm 0.00139}(4)=$ | $3.87\text{e-}10_{\pm 5.31\text{e-}10}(1)$ |
| mDTLZ2 | 3 | $0.0059_{\pm 0.000825}(1)+$ | $0.00641_{\pm 0.000891}(3)=$ | $0.00622_{\pm 0.000821}(2)=$ | $0.00983_{\pm 0.00271}(5)\text{-}$ | $0.00658_{\pm 0.00105}(4)$ |
| | 5 | $0.00461_{\pm 0.000966}(2)=$ | $0.00479_{\pm 0.000575}(4)=$ | $0.00458_{\pm 0.000553}(1)=$ | $0.0182_{\pm 0.0125}(5)\text{-}$ | $0.0047_{\pm 0.000487}(3)$ |
| | 8 | $0.00281_{\pm 0.00326}(1)=$ | $0.00578_{\pm 0.00696}(4)=$ | $0.00363_{\pm 0.00375}(3)=$ | $0.0282_{\pm 0.0228}(5)\text{-}$ | $0.00358_{\pm 0.00499}(2)$ |
| MaF2 | 3 | $0.00888_{\pm 1.68\text{e-}16}(4)=$ | $0.00887_{\pm 8.8\text{e-}05}(1)=$ | $0.00888_{\pm 1.22\text{e-}16}(3)=$ | $0.0089_{\pm 0.000103}(5)=$ | $0.00888_{\pm 1.15\text{e-}05}(2)$ |
| DTLZ5 | 3 | $2.78\text{e-}15_{\pm 7.93\text{e-}15}(1)+$ | $2.38\text{e-}13_{\pm 7.53\text{e-}13}(4)=$ | $4.6\text{e-}14_{\pm 2.41\text{e-}12}(2)+$ | $4.43\text{e-}13_{\pm 1.83\text{e-}12}(5)=$ | $1.43\text{e-}13_{\pm 6.6\text{e-}13}(3)$ |
| IMOP4 | 3 | $1.66\text{e-}17_{\pm 6.3\text{e-}17}(1)=$ | $1.94\text{e-}17_{\pm 7.46\text{e-}17}(2)=$ | $8.12\text{e-}17_{\pm 2.97\text{e-}16}(4)=$ | $0.000167_{\pm 0.000917}(5)=$ | $4.16\text{e-}17_{\pm 1.09\text{e-}16}(3)$ |
| IMOP6 | 3 | $3.66\text{e-}13_{\pm 1.69\text{e-}12}(2)\text{-}$ | $1.09\text{e-}12_{\pm 5.21\text{e-}12}(3)=$ | $2.16\text{e-}07_{\pm 1.18\text{e-}06}(4)\text{-}$ | $1.72\text{e-}05_{\pm 8.13\text{e-}05}(5)\text{-}$ | $3.2\text{e-}15_{\pm 8.49\text{e-}15}(1)$ |
| MP-DMP | 3 | $0.00778_{\pm 5.07\text{e-}05}(2)=$ | $0.00779_{\pm 7.8\text{e-}05}(3)=$ | $0.0078_{\pm 0.000217}(4)=$ | $0.051_{\pm 0.221}(5)=$ | $0.00777_{\pm 2.85\text{e-}05}(1)$ |
| | 5 | $0.0128_{\pm 0.000129}(4)=$ | $0.0127_{\pm 0.000116}(3)=$ | $0.0127_{\pm 0.000135}(2)=$ | $0.0387_{\pm 0.082}(5)=$ | $0.0127_{\pm 4.88\text{e-}05}(1)$ |
| ML-DMP | 3 | $0.0162_{\pm 7.43\text{e-}05}(3)=$ | $0.0161_{\pm 3.15\text{e-}05}(1)=$ | $0.0162_{\pm 0.000139}(5)=$ | $0.0162_{\pm 4.45\text{e-}05}(2)=$ | $0.0162_{\pm 8.56\text{e-}05}(4)$ |
| | 5 | $0.0452_{\pm 0.0563}(3)=$ | $0.0358_{\pm 0.09}(1)+$ | $0.0577_{\pm 0.116}(4)\text{-}$ | $0.0822_{\pm 0.171}(5)=$ | $0.0397_{\pm 0.072}(2)$ |
| RE3-4-7 | 3 | $0.0803_{\pm 0.00633}(4)=$ | $0.0798_{\pm 0.00868}(3)=$ | $0.0775_{\pm 0.01}(1)=$ | $0.0815_{\pm 0.014}(5)=$ | $0.0787_{\pm 0.00862}(2)$ |
| CRE5-3-1 | 5 | $0.191_{\pm 2.15\text{e-}08}(2)=$ | $0.191_{\pm 1.12\text{e-}08}(4)=$ | $0.191_{\pm 3.92\text{e-}08}(1)=$ | $0.191_{\pm 0.000559}(5)\text{-}$ | $0.191_{\pm 9.99\text{e-}09}(3)$ |
| Total +/=/- | | 2/25/1 | 1/26/1 | 2/19/7 | 2/13/13 | \ |
| Average rank | | 2.5714(2) | 2.6071(3) | 2.8571(4) | 4.4286(5) | 2.5357(1) |

## E.2  Impact of $\mu$ and $\tau_u$

We investigate the impact of parameters in BDNE. Recall that $\mu$ is the user-defined upper bound of the trade-off, and $\tau_u$ denotes the maximum number of iterations involved in the upper-level optimization.

The Pareto-optimal objective vector, situated on the boundary of a convex $PF$, usually has a substantially higher value of $M$ (see Definition 10) than that in the central region. We test BDNE with

different settings of $\mu$ on mDTLZ2, since this problem has a convex $PF$ and thus effectively discloses the impact of $\mu$. We consider 6 values for $\mu$: 1, 20, 40, 60, 80, and 100. We find that the estimated nadir objective vector is dominated by the exact one on every instance. We introduce another metric, representing the distance between the estimated nadir objective vector and the ideal objective vector. This metric is defined as $E' = \sqrt{\sum_{i=1}^{m} \left( (\hat{z}_i^{nad} - z_i^{ide})/(z_i^{nad} - z_i^{ide}) \right)^2}$. Figure 7 further illustrate this observation, plotting the means of $E$ and $E'$. Consistently, the obtained nadir objective vector deviates from the exact nadir objective vector and moves closer to the ideal objective vector, as the value of $\mu$ decreases. A smaller $\mu$ implies that decision-makers have a stricter requirement for preferred solutions, resulting in a smaller set of preferred solutions. Accordingly, the promising region becomes more refined and the estimated nadir objective vector is getting farther away from the exact one.

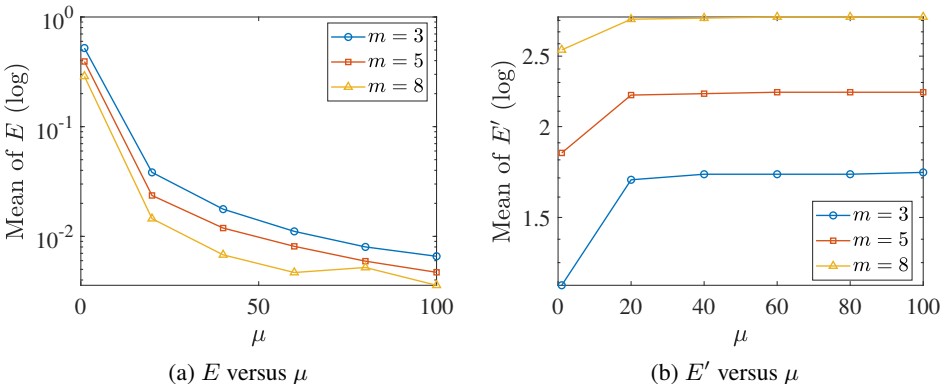

(a) $E$ versus $\mu$       (b) $E'$ versus $\mu$

Figure 7: Results of different $\mu$ settings on mDTLZ2.

Different MOPs have different difficulties in estimating the nadir objective vector. To clearly show the effects of $\tau_u$, we employ TN4, which possesses a complicated feasible objective region. Figure 8 plots the results. The value for $\tau_u$ begins at 2, increases by 2, and ends at 18, while $\tau_l$ remains 200. The runtime is getting shorter as the value of $\tau_u$ decreases, which is straightforward. Besides, the relationship between the runtime and $\tau_u$ is linear. The curve of $E$ versus $\tau_u$ exhibits a declining trend with intermittent fluctuations. Generally, the error is getting smaller (or the accuracy is getting higher) as the value of $\tau_u$ increases. The fluctuations occur because the lower-level optimization algorithm is stochastic (*i.e.*, it does not guarantee the attainment of a Pareto-optimal solution within a finite number of iterations).

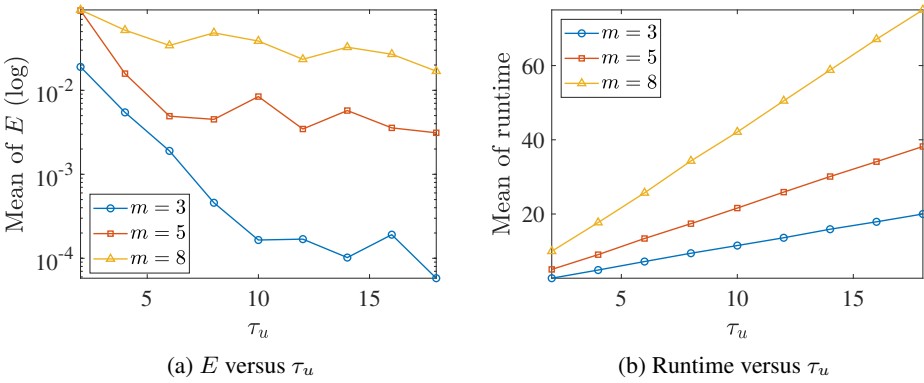

(a) $E$ versus $\tau_u$       (b) Runtime versus $\tau_u$

Figure 8: Results of different $\tau_u$ settings on TN4.

## E.3 Comparison with Other Heuristic Methods

In previous sections, we employ theoretical analysis, illustrative examples, and experimental comparisons with two representative baseline methods to demonstrate the advantage of BDNE over the existing heuristic ones. To more comprehensively validate the effectiveness of BDNE, we present the experimental results comparing our method with the remaining heuristic methods in Table 8. The majority of heuristic methods cannot surpass our method on any instance, while our method significantly outperforms them on most instances. Table 9 shows the detailed results of the methods having better performance than BDNE on some instances. Although EP1 achieves lower errors on 9 instances, the gaps are relatively small ($\Delta < 0.03$). In contrast, BDNE has remarkably better performance than EP1 on many instances. For example, $\Delta < -0.3$ on TN4 and TN4; $\Delta < -1.5$ on 5- and 8-objective DTLZ2; $\Delta < -12$ on 5-objective ML-DMP. For EP4, the better result only appears on 3-objective ML-DMP, and the gap is very small ($\Delta < 10^{-4}$). EP4 shows poor results on the rest of the instances. We can conclude that our method still outperforms other heuristic methods.

Table 8: Overall comparisons of error metric values between other heuristic methods and BDNE.

|  | SF2 | SF3 | SF4 | EP1 | EP2 | EP3 | EP4 | EP5 | BDNE |
|---|---|---|---|---|---|---|---|---|---|
| Total +/=/- | 0/0/28 | 0/0/28 | 0/0/28 | 9/3/16 | 0/0/28 | 0/2/26 | 1/2/25 | 0/1/27 | \ |
| Average rank | 6.1786(6) | 6.4286(7) | 4.4286(3) | 2.7143(2) | 7.9286(9) | 4.6429(5) | 4.4643(4) | 6.75(8) | 1.4643(1) |

Table 9: Comparisons of error metric values among EP1, EP4, and BDNE.

| Problem | $m$ | EP1 | $\Delta$ | EP4 | $\Delta$ | BDNE |
|---|---|---|---|---|---|---|
| TN1 | 3 | $1.71e\text{-}15_{\pm 3.33e\text{-}15}(2)$= | -1.39e-15 | $0.373_{\pm 0.39}(6)$- | -0.373 | $3.14e\text{-}16_{\pm 6.65e\text{-}16}(1)$ |
|  | 5 | $4.32e\text{-}17_{\pm 2.04e\text{-}16}(1)$+ | 0.000273 | $0.263_{\pm 0.152}(5)$- | -0.263 | $0.000273_{\pm 0.000741}(2)$ |
|  | 8 | $4.44e\text{-}17_{\pm 2.05e\text{-}16}(1)$+ | 0.0221 | $0.227_{\pm 0.106}(3)$- | -0.205 | $0.0221_{\pm 0.0331}(2)$ |
| TN2 | 3 | $0.333_{\pm 0}(4)$- | -0.333 | $0.437_{\pm 0.246}(6)$- | -0.437 | $0.000172_{\pm 0.000535}(1)$ |
|  | 5 | $0.333_{\pm 0}(3)$- | -0.325 | $0.454_{\pm 0.169}(6)$- | -0.446 | $0.00803_{\pm 0.0233}(1)$ |
|  | 8 | $0.336_{\pm 0.0151}(2)$- | -0.318 | $0.36_{\pm 0.0728}(4)$- | -0.341 | $0.0185_{\pm 0.0204}(1)$ |
| TN3 | 3 | $6.39e\text{-}16_{\pm 9.85e\text{-}16}(2)$= | -4e-16 | $0.274_{\pm 0.23}(6)$- | -0.274 | $2.39e\text{-}16_{\pm 2.32e\text{-}16}(1)$ |
|  | 5 | $5.28e\text{-}17_{\pm 1.73e\text{-}16}(1)$+ | 0.000669 | $0.221_{\pm 0.115}(4)$- | -0.22 | $0.000669_{\pm 0.00203}(2)$ |
|  | 8 | $4.44e\text{-}17_{\pm 1.69e\text{-}16}(1)$+ | 0.0195 | $0.258_{\pm 0.175}(3)$- | -0.238 | $0.0195_{\pm 0.0263}(2)$ |
| TN4 | 3 | $0.333_{\pm 0}(4)$- | -0.333 | $0.409_{\pm 0.14}(6)$- | -0.409 | $4.92e\text{-}05_{\pm 5.59e\text{-}05}(1)$ |
|  | 5 | $0.333_{\pm 0}(3)$- | -0.331 | $0.467_{\pm 0.281}(6)$- | -0.465 | $0.00208_{\pm 0.00422}(1)$ |
|  | 8 | $0.333_{\pm 6.39e\text{-}16}(2)$- | -0.306 | $0.382_{\pm 0.097}(4)$- | -0.356 | $0.0268_{\pm 0.031}(1)$ |
| DTLZ2 | 3 | $4.55e\text{-}12_{\pm 1.06e\text{-}11}(1)$+ | 1.77e-10 | $2.33e\text{-}06_{\pm 1.16e\text{-}05}(4)$- | -2.33e-06 | $1.81e\text{-}10_{\pm 2.82e\text{-}10}(2)$ |
|  | 5 | $1.67_{\pm 0.138}(7)$- | -1.67 | $0.00127_{\pm 0.00241}(3)$= | -0.00127 | $1.68e\text{-}10_{\pm 1.86e\text{-}10}(1)$ |
|  | 8 | $2.38_{\pm 0.101}(7)$- | -2.38 | $0.0242_{\pm 0.0147}(3)$- | -0.0242 | $3.87e\text{-}10_{\pm 5.31e\text{-}10}(1)$ |
| mDTLZ2 | 3 | $0.000623_{\pm 0.00198}(1)$+ | 0.00596 | $0.0405_{\pm 0.0439}(5)$- | -0.0339 | $0.00658_{\pm 0.00105}(2)$ |
|  | 5 | $0.00155_{\pm 0.00409}(1)$+ | 0.00314 | $0.044_{\pm 0.0261}(4)$- | -0.0393 | $0.0047_{\pm 0.000487}(2)$ |
|  | 8 | $0.00279_{\pm 0.0105}(1)$+ | 0.000798 | $0.0387_{\pm 0.0295}(3)$- | -0.0352 | $0.00358_{\pm 0.00499}(2)$ |
| MaF2 | 3 | $0.643_{\pm 0.215}(6)$- | -0.635 | $0.0147_{\pm 0.0105}(2)$= | -0.00577 | $0.00888_{\pm 1.15e\text{-}05}(1)$ |
| DTLZ5 | 3 | $3.23e\text{-}10_{\pm 2.71e\text{-}10}(2)$- | -3.23e-10 | $0.582_{\pm 0.556}(4)$- | -0.582 | $1.43e\text{-}13_{\pm 6.6e\text{-}13}(1)$ |
| IMOP4 | 3 | $8.74e\text{-}12_{\pm 2.19e\text{-}11}(2)$- | -8.74e-12 | $0.949_{\pm 0.00159}(6)$- | -0.949 | $4.16e\text{-}17_{\pm 1.09e\text{-}16}(1)$ |
| IMOP6 | 3 | $5.77e\text{-}07_{\pm 3.16e\text{-}06}(2)$- | -5.77e-07 | $0.13_{\pm 0.337}(5)$- | -0.13 | $3.2e\text{-}15_{\pm 8.49e\text{-}15}(1)$ |
| MP-DMP | 3 | $0.00776_{\pm 8.4e\text{-}06}(1)$= | 1.13e-05 | $0.219_{\pm 0.0119}(7)$- | -0.211 | $0.00777_{\pm 2.85e\text{-}05}(2)$ |
|  | 5 | $0.0167_{\pm 0.0218}(2)$- | -0.00405 | $0.849_{\pm 0.0715}(6)$- | -0.836 | $0.0127_{\pm 4.88e\text{-}05}(1)$ |
| ML-DMP | 3 | $1.21_{\pm 1.04}(7)$- | -1.19 | $0.0161_{\pm 3.28e\text{-}05}(1)$+ | 3.9e-05 | $0.0162_{\pm 8.56e\text{-}05}(4)$ |
|  | 5 | $12.9_{\pm 11}(7)$- | -12.8 | $0.547_{\pm 0.0867}(2)$- | -0.507 | $0.0397_{\pm 0.072}(1)$ |
| RE3-4-7 | 3 | $0.148_{\pm 3.47e\text{-}06}(2)$- | -0.0692 | $0.616_{\pm 0.0117}(6)$- | -0.537 | $0.0787_{\pm 0.00862}(1)$ |
| CRE5-3-1 | 5 | $0.19_{\pm 0.00396}(1)$+ | 0.000722 | $1.25_{\pm 0.0455}(5)$- | -1.06 | $0.191_{\pm 9.99e\text{-}09}(2)$ |
| Total +/=/- |  | 9/3/16 |  | 1/2/25 |  | \ |
| Average rank |  | 2.7143(2) |  | 4.4643(4) |  | 1.4643(1) |

### E.4 Pilot Studies on Integrating BDNE in Algorithm Iteration

### E.4 Pilot Studies on Integrating BDNE in Algorithm Iteration

We present BDNE as an independent algorithm in this paper. This subsection provides initial investigations about integrating BDNE into the iteration of an algorithm. We select the MOEA to conduct experimental studies and propose two simple implementations. First, we replace the nadir objective vector estimation method of an MOEA with ours and denote the variant as "V1". We define the promising region as the dominating region of the estimated nadir objective vector. Then V1 is further improved by focusing the search within the promising region. Specifically, in the environment selection of each iteration, the objective vector outside the promising region is considered to have a worse fitness value than the one inside. The improved version of V1 is termed "V2". We consider two famous MOEAs: MOEA/D [42] and NSGA-III [32]. The population size of the BDNE is set to 12, whereas the population size of the MOEA is set to 91. The number of function evaluations is consistent with the original settings, *i.e.*, 180,000. The MOEA and its improved versions use the same function evaluation budget. V1 or V2 consumes 12 additional function evaluations for nadir objective vector estimation in each iteration, and as a result, it uses about 230 fewer iterations than the original MOEA.

The statistical results of the hypervolume metric [61] are reported in Table 10. The results of Wilcoxon's rank sum test are shown by two symbols in $\{+,=,-\}$ in the cell. The first and second symbols are the comparison results with V1 and V2, respectively. We begin by analyzing the comparative results of the MOEA and V1. Both MOEA/D-V1 and NSGA-III-V1 significantly outperform their original versions, respectively. Besides, MOEA/D-V1 exhibits clearly smaller standard deviations compared with its original versions (*e.g.*, 0.0106 versus 0.00289), indicating more stable performance. Analogously, NSGA-III-V1 yields stable performance. These results also emphasize that inaccurate nadir objective vector estimation can have a significant impact on algorithm performance. According to the remaining results, V2 of the MOEA has a better performance than V1, indicating the promising region determined by BDNE is useful for guiding the search. Furthermore, we find that V2, on average, retains more solutions within the promising region compared with V1. For MOEA/D, 73 versus 83 on TN1, 46 versus 53 on TN2, 70 versus 84 on TN3, and 48 versus 53 on TN4. For NSGA-III, 30 versus 89 on TN1, 25 versus 67 on TN2, 32 versus 88 on TN3, and 25 versus 66 on TN4. This explains the performance improvement and demonstrates the effectiveness of the scheme.

In conclusion, our method shows great potential to improve the performance of an algorithm.

Table 10: Comparisons of hypervolume metric values between the MOEA and its improved variants.

| Problem | MOEA/D | V1 | V2 | NSGA-III | V1 | V2 |
|---|---|---|---|---|---|---|
| TN1 | $0.387_{\pm 0.0106}(3)\text{- -}$ | $0.398_{\pm 0.00289}(2)\backslash\text{-}$ | $0.404_{\pm 0.00369}(1)+\backslash$ | $0.333_{\pm 0.0209}(3)\text{- -}$ | $0.377_{\pm 0.00331}(2)\backslash\text{-}$ | $0.406_{\pm 0.00243}(1)+\backslash$ |
| TN2 | $0.213_{\pm 0.0154}(3)\text{- -}$ | $0.227_{\pm 0.00234}(1)\backslash+$ | $0.226_{\pm 0.0217}(2)\text{-}\backslash$ | $0.206_{\pm 0.0166}(3)=\text{-}$ | $0.207_{\pm 0.00261}(2)\backslash\text{-}$ | $0.23_{\pm 0.00169}(1)+\backslash$ |
| TN3 | $0.384_{\pm 0.0135}(3)\text{- -}$ | $0.397_{\pm 0.00475}(2)\backslash\text{-}$ | $0.404_{\pm 0.00313}(1)+\backslash$ | $0.34_{\pm 0.0271}(3)\text{- -}$ | $0.378_{\pm 0.00355}(2)\backslash\text{-}$ | $0.405_{\pm 0.00176}(1)+\backslash$ |
| TN4 | $0.209_{\pm 0.022}(3)\text{- -}$ | $0.227_{\pm 0.00234}(2)\backslash\text{-}$ | $0.229_{\pm 0.00145}(1)+\backslash$ | $0.148_{\pm 0.0298}(3)\text{- -}$ | $0.209_{\pm 0.00308}(2)\backslash\text{-}$ | $0.229_{\pm 0.00212}(1)+\backslash$ |
| Total +/=/- | 0/0/4 0/0/4 | \ 1/0/3 | 3/0/1 \ | 0/1/3 0/0/4 | \ 0/0/4 | 4/0/0 \ |
| Average rank | 3(3) | 1.75(2) | 1.25(1) | 3(3) | 2(2) | 1(1) |

### E.5 Results on All Problems in the Used Test Suites

In previous experiments, we select the problems with different kinds of feasible objective regions from several test suites to comprehensively compare the algorithms and avoid showing similar results. Specifically, we employ 4 new and 6 existing test problems (28 instances in total), including instances with many objectives (*e.g.*, 5 and 8 objectives cases), weakly Pareto-optimal boundaries (*e.g.*, TN1-TN4 and mDTLZ3), linear $PF$s (*e.g.*, TN1 and TN3), convex $PF$s (*e.g.*, mDTLZ3), concave $PF$s (*e.g.*, TN2, TN4, and DTLZ3), irregular $PF$s (*e.g.*, TN3, TN4, DTLZ5, IMOP4, and IMOP6).

Table 11 summarizes the complete results on all the test problems by showing "Total +/=/-". DTLZ, MP-DMP, and ML-DMP are absent in this experiment, as the MaF test suite already covers them. Besides, IMOP1-IMOP3 are non-scalable 2-objective MOPs, which are also omitted. We can find that our method outperforms the two representative algorithms on most instances. Table 12 records the detailed results indicating either of the two algorithms statistically outperforms BDNE, and the positive gaps in the table are marked. Most of the positive gaps are lower than $0.02$, while the absolute

values of negative gaps can be very large. Besides, BDNE always ranks second on these instances, exhibiting competitive performance. In short, BDNE still shows its superiority on the three test suites.

Table 11: Overall comparisons of error metric values among ECR-NSGA-II, DNPE, and BDNE on MaF, mDTLZ, and IMOP test suites.

| Problem | # instances | EC-NSGA-II | DNPE | BDNE |
|---------|-------------|------------|------|------|
| MaF1-MaF7 | 21 | 1/0/20 | 5/2/14 | \ |
| MaF8-MaF9 | 6 | 0/0/6 | 1/0/5 | \ |
| MaF10-MaF13 | 12 | 5/0/7 | 0/1/11 | \ |
| mDTLZ1-mDTLZ4 | 12 | 0/0/12 | 1/2/9 | \ |
| IMOP4-IMOP8 | 5 | 0/0/5 | 0/1/4 | \ |
| Average rank | | 2.5893(3) | 2.125(2) | 1.2857(1) |

Table 12: Comparisons of error metric values among ECR-NSGA-II, DNPE, and BDNE on the three test suites' instances where either of the two algorithms outperform BDNE.

| Problem | $m$ | ECR-NSGA-II | $\Delta$ | DNPE | $\Delta$ | BDNE |
|---------|-----|-------------|----------|------|----------|------|
| MaF1 | 8 | $0.2_{\pm 0.067}(3)-$ | -0.2 | $2.84\text{e-}06_{\pm 6.48\text{e-}09}(1)+$ | 0.000149 | $0.000151_{\pm 0.000292}(2)$ |
| MaF2 | 8 | $0.275_{\pm 0.00607}(3)-$ | -0.0805 | $1.69\text{e-}06_{\pm 3.75\text{e-}11}(1)+$ | 0.195 | $0.195_{\pm 0.0551}(2)$ |
| MaF4 | 3 | $0.00569_{\pm 0.0168}(1)+$ | 0.000713 | $0.0127_{\pm 0.000421}(3)-$ | -0.00627 | $0.0064_{\pm 0.000955}(2)$ |
| MaF4 | 5 | $0.133_{\pm 0.369}(3)-$ | -0.129 | $0.00255_{\pm 0.000427}(1)+$ | 0.000828 | $0.00337_{\pm 0.000761}(2)$ |
| MaF4 | 8 | $0.365_{\pm 0.342}(3)-$ | -0.364 | $0.000355_{\pm 0.000114}(1)+$ | 0.00126 | $0.00161_{\pm 0.000769}(2)$ |
| MaF7 | 3 | $0.166_{\pm 0.252}(3)-$ | -0.0043 | $0.0269_{\pm 0.129}(1)+$ | 0.135 | $0.162_{\pm 0.334}(2)$ |
| MaF10 | 3 | $0.00669_{\pm 7.58\text{e-}08}(1)+$ | 0.00674 | $0.14_{\pm 6.38\text{e-}07}(3)-$ | -0.127 | $0.0134_{\pm 0.000161}(2)$ |
| MaF10 | 5 | $0.0148_{\pm 8.51\text{e-}08}(1)+$ | 0.0154 | $0.352_{\pm 1.35\text{e-}06}(3)-$ | -0.322 | $0.0302_{\pm 0.00333}(2)$ |
| MaF10 | 8 | $0.0298_{\pm 6.23\text{e-}08}(1)+$ | 0.0156 | $0.694_{\pm 0.0838}(3)-$ | -0.648 | $0.0454_{\pm 0.011}(2)$ |
| MaF11 | 5 | $0.0165_{\pm 0.0175}(1)+$ | 0.0103 | $0.847_{\pm 0.00287}(3)-$ | -0.82 | $0.0268_{\pm 0.00371}(2)$ |
| MaF11 | 8 | $0.0256_{\pm 0.000378}(1)+$ | 0.051 | $1.64_{\pm 0.00821}(3)-$ | -1.56 | $0.0766_{\pm 0.0145}(2)$ |
| mDTLZ1 | 8 | $711_{\pm 6.65}(3)-$ | -711 | $9.36\text{e-}05_{\pm 0.000104}(1)+$ | 0.0363 | $0.0364_{\pm 0.197}(2)$ |
| ML-DMP | 3 | $80.8_{\pm 0.0603}(3)-$ | -80.7 | $0.00699_{\pm 2.82\text{e-}05}(1)+$ | 0.00917 | $0.0162_{\pm 8.56\text{e-}05}(2)$ |

# F  Proofs

## F.1  Properties of Cone Domination

**Property 1** (Reflexivity). $\forall \mathbf{u}, \mathbf{u} \not\prec^c \mathbf{u}$.

*Proof.* Since $\tilde{u}_i = \tilde{u}_i$ for $i = 1, \ldots, m$, we have $\mathbf{u} \not\prec^c \mathbf{u}$. $\qquad\square$

**Property 2** (Anti-symmetry). *If* $\mathbf{u} \prec^c \mathbf{v}$, *then* $\mathbf{v} \not\prec^c \mathbf{u}$.

*Proof.* Since $\mathbf{u} \prec^c \mathbf{v}$, we have $\tilde{\mathbf{u}} \prec \tilde{\mathbf{v}}$. Thus, $\mathbf{v} \not\prec^c \mathbf{u}$ according to Definition 8. $\qquad\square$

**Property 3** (Transitivity). *If* $\mathbf{t} \prec^c \mathbf{u}$ *and* $\mathbf{u} \prec^c \mathbf{v}$, *then* $\mathbf{t} \prec^c \mathbf{v}$.

*Proof.* Since $\mathbf{t} \prec^c \mathbf{u}$ and $\mathbf{u} \prec^c \mathbf{v}$, we have $\tilde{\mathbf{t}} \prec \tilde{\mathbf{u}}$ and $\tilde{\mathbf{u}} \prec \tilde{\mathbf{v}}$. According to the transitivity of Pareto domination, $\tilde{\mathbf{t}} \prec \tilde{\mathbf{v}}$. Therefore, $\mathbf{t} \prec^c \mathbf{v}$. $\qquad\square$

The three properties imply that cone domination defines a strict partial order.

## F.2  Proof of Theorem 1

*Proof.* Without loss of generality, we let $z_j^r = 0$ for $j = 1, \ldots, m$.

(Sufficiency) Let $z_j^* \geq z_j^r = 0$ for at least one $j \in \{1, \ldots, m\} \setminus \{i\}$. We consider an objective vector $\mathbf{z}$ satisfying $\mathbf{z}^* \prec \mathbf{z}$. We have $\sum_{k=1}^m z_k > \sum_{k=1}^m z_k^*$ and then

$$(1-\alpha)z_j + \frac{\alpha}{m}\sum_{k=1}^m z_k > (1-\alpha)z_j^* + \frac{\alpha}{m}\sum_{k=1}^m z_k^* \quad \text{for every} \quad j \in \{1, \ldots, m\}. \tag{23}$$

Consequently,

$$\begin{aligned} w_j^i \cdot \tilde{z}_j = w_j^i \cdot \tilde{z}_j^* = 0, & \quad \text{if } w_j^i = 0, \\ w_j^i \cdot \tilde{z}_j > w_j^i \cdot \tilde{z}_j^*, & \quad \text{if } w_j^i > 0. \end{aligned} \tag{24}$$

Since $\exists j \in \{1, \ldots, m\} \setminus \{i\}$ such that $\tilde{z}_j^* \geq 0$, we can deduce

$$\max_{1 \leq j \leq m} \left\{ w_j^i \cdot \tilde{z}_j \right\} > \max_{1 \leq j \leq m} \left\{ w_j^i \cdot \tilde{z}_j^* \right\} \geq 0. \tag{25}$$

The dominated objective vector can not be optimal for the boundary subproblem. Therefore, the optimal solution of the boundary subproblem must be Pareto-optimal.

(Necessity) Let $\mathbf{z}^*$ be Pareto-optimal. We suppose that $z_j^r > \tilde{z}_j^*$ for each $j \in \{1, \ldots, m\} \setminus \{i\}$. We consider $z_j = z_j^* + \epsilon_j$ for $j = 1, \ldots, m$ where $\epsilon_j \geq 0$ is a sufficiently small value such that $\tilde{z}_j^* < \tilde{z}_j < z_j^r = 0$. Then the following equation holds

$$\max_{1 \leq j \leq m} \left\{ w_j^i \cdot \tilde{z}_j^* \right\} = \max_{1 \leq j \leq m} \left\{ w_j^i \cdot \tilde{z}_j \right\} = 0, \tag{26}$$

which indicates that $\mathbf{z}$ dominated by $\mathbf{z}^*$ can also be optimal for the boundary subproblem. But this is a contradiction. $\qquad\square$

## F.3  Proof of Theorem 2

*Proof.* Without loss of generality, we let $z_j^r = 0$ for $j = 1, \ldots, m$. Since $\tilde{z}_j^{(i)^c} \geq z_j^r$ for every $j \in \{1, \ldots, m\} \setminus \{i\}$, we have

$$\max_{1 \leq j \leq m} \left\{ w_j^i \cdot \tilde{z}_j^{(i)^c} \right\} = w_j^i \cdot \tilde{z}_j^{(i)^c} \ (j \neq i) \geq 0. \tag{27}$$

(Sufficiency) Let $\mathbf{z}^{(i)^c}$ be cone-optimal. We assume that $\mathbf{z}$, rather than $\mathbf{z}^{(i)^c}$, is the optimal objective vector to this boundary subproblem. That is,

$$\max_{1 \leq j \leq m} \left\{ w_j^i \cdot \tilde{z}_j \right\} < \max_{1 \leq j \leq m} \left\{ w_j^i \cdot \tilde{z}_j^{(i)^c} \right\}. \tag{28}$$

Since $\mathbf{z} \not\prec^c \mathbf{z}^{(i)^c}$ and $w_i^i \cdot \tilde{z}_i^{(i)^c} = 0$, Eq. (28) implies that $\tilde{z}_i^{(i)^c} < \tilde{z}_i$ and $\tilde{z}_j^{(i)^c} > \tilde{z}_j$ for all $j \in \{1, \dots, m\} \setminus \{i\}$. Theorem 1 signifies $\mathbf{z}$ is Pareto-optimal, and thus the corresponding objective function value is larger than that of the critical point (i.e., $z_i^{(i)^c} \geq z_i$). Based on $\tilde{z}_i^{(i)^c} < \tilde{z}_i$ and $z_i^{(i)^c} \geq z_i$, we have

$$\sum_{j=1}^{m} z_j^{(i)^c} < \sum_{j=1}^{m} z_j. \tag{29}$$

There exists an index $l \neq i$ such that $z_l^{(i)^c} < z_l$. Therefore,

$$z_l^{(i)^c} + \frac{\alpha}{m} \sum_{j=1}^{m} z_l^{(i)^c} < z_l + \frac{\alpha}{m} \sum_{j=1}^{m} z_l, \tag{30}$$

which means

$$\max_{1 \leq j \leq m} \left\{ w_j^i \cdot \tilde{z}_j \right\} > w_l^i \cdot \tilde{z}_l^{(i)^c} = \max_{1 \leq j \leq m} \left\{ w_j^i \cdot \tilde{z}_j^{(i)^c} \right\}. \tag{31}$$

This conclusion conflicts with the assumption. $\mathbf{z}^{(i)^c}$ is optimal for this boundary subproblem. Since $\mathbf{z}$ can be arbitrary, $\mathbf{z}^{(i)^c}$ is also the unique optimal objective vector.

(Necessity) Let $\mathbf{z}^{(i)^c}$ uniquely solve the boundary subproblem with

$$w_j^i = \begin{cases} 0, & j = i, \\ \dfrac{\beta}{\tilde{z}_j^{(i)^c} - z_j^r}, & j \neq i. \end{cases} \tag{32}$$

We assume that $\mathbf{z}$ cone-dominates $\mathbf{z}^{(i)^c}$. Then

$$\begin{aligned} w_j^i \cdot \tilde{z}_j = w_j^i \cdot \tilde{z}_j^{(i)^c} = 0, & \quad \text{if } w_j^i = 0, \\ w_j^i \cdot \tilde{z}_j \leq w_j^i \cdot \tilde{z}_j^{(i)^c}, & \quad \text{if } w_j^i > 0. \end{aligned} \tag{33}$$

Consequently,

$$\max_{1 \leq j \leq m} \left\{ w_j^i \cdot \tilde{z}_j \right\} \leq \max_{1 \leq j \leq m} \left\{ w_j^i \cdot \tilde{z}_j^{(i)^c} \right\}, \tag{34}$$

indicating that $\mathbf{z}^{(i)^c}$ is not uniquely optimal for the boundary subproblem. But this is a contradiction. □

## F.4 Proof of Corollary 1

*Proof.* Since the index $l$ in Inequality (30) is not arbitrary, $z_j^{(i)^c}$ should be larger than $z_j^r$ for $j = 1, \dots, m$. □

## F.5 Proof of Theorem 3

*Proof.* We first prove that a cone-optimal objective vector is Pareto-optimal. Letting $\mathbf{z}^*$ be a cone-optimal objective vector and $\mathbf{z} \in \{\mathbf{f}(\mathbf{x}) | \mathbf{x} \in \Omega\}$, we have

$$\exists k, (1 - \alpha)(z_k^* - z_k) + \frac{\alpha}{m} \left( \sum_{j=1}^{m} z_j^* - \sum_{j=1}^{m} z_j \right) < 0. \tag{35}$$

Since

$$\min_{1 \leq j \leq m} (z_j^* - z_j) < \frac{1}{m} \sum_{j=1}^{m} (z_j^* - z_j), \tag{36}$$

then $\exists k, z_k^* - z_k < 0$.

We now prove that $\alpha \leq \frac{m}{(m-1)M+m}$ holds for a cone-optimal objective vector. Given a Pareto-optimal objective vector $\mathbf{z}^*$ and an objective vector $\mathbf{z} \in \{\mathbf{f}(\mathbf{x})|\mathbf{x} \in \Omega\}$, to make sure that $\tilde{\mathbf{z}}^*$ is Pareto-optimal for each $\tilde{\mathbf{z}}$, the following inequality should hold

$$
\min_{1 \leq j \leq m} \tilde{z}_j^* - \tilde{z}_j
$$

$$
=(1-\alpha) \min_{1 \leq j \leq m} (z_j^* - z_j) + \frac{\alpha}{m} \left( \sum_{j=1}^{m} z_j^* - \sum_{j=1}^{m} z_j \right) \tag{37}
$$

$$
<0.
$$

Since $\exists j, z_j^* - z_j < 0$, we have

$$
\sum_{j=1}^{m} \left( z_j^* - z_j \right) < (m-1) \max_{1 \leq j \leq m} \left( z_j^* - z_j \right). \tag{38}
$$

The Definition 10 is equivalent to a properly Pareto-optimal decision vector $\mathbf{x}^* \in \Omega$ satisfying that

$$
\frac{\max_{1 \leq j \leq m} f_j(\mathbf{x}^*) - f_j(\mathbf{x})}{\max_{1 \leq j \leq m} f_j(\mathbf{x}) - f_j(\mathbf{x}^*)} \leq M < \infty \tag{39}
$$

where $\mathbf{x} \in \Omega$ and $\max_{1 \leq j \leq m} f_j(\mathbf{x}) - f_j(\mathbf{x}^*) > 0$. According to Inequalities (38) and (39), we have

$$
\frac{\alpha}{m} \left( \sum_{j=1}^{m} z_j^* - \sum_{j=1}^{m} z_j \right) < \frac{(m-1)\alpha}{m} \max_{1 \leq j \leq m} \left( z_j^* - z_j \right) \leq \frac{(m-1)\alpha M}{m} \max_{1 \leq j \leq m} \left( z_j - z_j^* \right). \tag{40}
$$

Then

$$
(1-\alpha) \min_{1 \leq j \leq m} (z_j^* - z_j) + \frac{\alpha}{m} \left( \sum_{j=1}^{m} z_j^* - \sum_{j=1}^{m} z_j \right)
$$

$$
< -(1-\alpha) \max_{1 \leq j \leq m} (z_j - z_j^*) + \frac{(m-1)\alpha M}{m} \max_{1 \leq j \leq m} (z_j - z_j^*) \tag{41}
$$

$$
\leq 0.
$$

Since $\max_{1 \leq j \leq m}(z_j - z_j^*) > 0$, then

$$
\alpha - 1 + \frac{(m-1)\alpha M}{m} \leq 0,
$$

$$
\alpha \leq \frac{m}{(m-1)M+m}. \tag{42}
$$

$\square$

### F.6 Proof of Corollary 2

If a cone-optimal objective vector is not properly Pareto-optimal, $\alpha \leq 0$ should hold according to Theorem 3. Therefore, a cone-optimal objective vector has a finite value of $M$, indicating it is properly Pareto-optimal. However, $\alpha > 0$ as shown in Definition 8. The formal proof is as follows.

*Proof.* Let $\mathbf{z}^*$ be a cone-optimal objective vector. We assume that $\mathbf{z}^*$ is not properly Pareto-optimal. Then there exists an objective vector $\mathbf{z}$ such that

$$
\frac{\max_{1 \leq j \leq m} z_j^* - z_j}{\max_{1 \leq j \leq m} z_j - z_j^*} \to \infty. \tag{43}
$$

Since $\mathbf{z}^*$ is a Pareto-optimal objective vector, we have

$$
\sum_{j=1}^{m} \left( z_j^* - z_j \right)
$$

$$
\geq \max_{1 \leq j \leq m} \left( z_j^* - z_j \right) - (m-1) \max_{1 \leq j \leq m} \left( z_j - z_j^* \right) \tag{44}
$$

$$
>0.
$$

The inequality can be rewritten as

$$(1 - \alpha) \min_{1 \leq j \leq m} \left( z_j^* - z_j \right) + \frac{\alpha}{m} \sum_{j=1}^{m} \left( z_j^* - z_j \right)$$

$$\geq \frac{\alpha}{m} \max_{1 \leq j \leq m} \left( z_j^* - z_j \right) - $$

$$\left( \frac{\alpha(m-1)}{m} + 1 - \alpha \right) \max_{1 \leq j \leq m} \left( z_j - z_j^* \right) \tag{45}$$

$$> 0.$$

This conclusion conflicts with the optimality of $\mathbf{z}^*$. $\qquad \square$

Moreover, leveraging Theorem 3 and Corollary 2, we illustrate the relationships among Pareto optimality, proper Pareto optimality, and cone optimality in Figure 9.

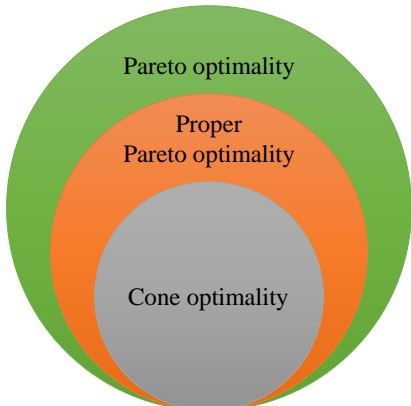

Figure 9: Relationships among Pareto optimality, proper Pareto optimality, and cone optimality.

### F.7 Proof of Theorem 4

The proof is analogous to the sufficiency proof of Theorem 2.

*Proof.* Without loss of generality, we let $z_j^r = 0$ for $j = 1, \ldots, m$. According to Theorem 6, we can let $w_j^i = \beta / \tilde{z}_j^*$ for every $j \in \{1, \ldots, m\} \setminus \{i\}$. Then we have

$$\max_{1 \leq j \leq m} \left\{ w_j^i \cdot \tilde{z}_j^* \right\} = \beta \geq 0, \tag{46}$$

where $\beta$ is a constant.

Suppose that $\mathbf{z}'$ yields a lower function value than $\mathbf{z}^*$ for the boundary subproblem with $\mathbf{w}^i$. That is,

$$\max_{1 \leq j \leq m} \left\{ w_j^i \cdot \tilde{z}_j' \right\} < \beta, \tag{47}$$

and $\mathbf{z}^* \not\prec \mathbf{z}'$ according to Theorem 1. Since $\mathbf{z}' \not\prec^c \mathbf{z}^*$, Eq. (47) implies that $\tilde{z}_i^* < \tilde{z}_i'$ and $\tilde{z}_j^* > \tilde{z}_j'$ for every $j \in \{1, \ldots, m\} \setminus \{i\}$. Based on $\tilde{z}_i^* < \tilde{z}_i'$ and $z_i^* \geq z_i'$, we can deduce that

$$\sum_{j=1}^{m} z_j^* < \sum_{j=1}^{m} z_j'. \tag{48}$$

There exists an index $l \neq i$ such that $z_l^* < z_l'$. Therefore,

$$z_l^* + \frac{\alpha}{m} \sum_{j=1}^{m} z_l^* < z_l' + \frac{\alpha}{m} \sum_{j=1}^{m} z_l', \tag{49}$$

which means

$$\max_{1 \leq j \leq m} \left\{ w_j^i \cdot \tilde{z}_j' \right\} \geq w_l^i \cdot \tilde{z}_l' > \beta. \tag{50}$$

This conclusion conflicts with the assumption. $\qquad \square$

## F.8   Proof of Corollary 3

*Proof.* According to Theorem 3, the Pareto-optimal objective vector deviating from the user-defined trade-off is cone-dominated by some objective vector. And Theorem 4 reports the cone-optimal objective vector that maximizes a given objective is optimal for at least one boundary subproblem.

$\square$

## F.9   Proof of Theorem 5

The proof is modified from the sufficiency proof of Theorem 1.

*Proof.* Without loss of generality, we let $z_j^r = 0$ for $j = 1, \ldots, m$. We consider an objective vector $\mathbf{z}$ satisfying $\mathbf{z}^* \prec^c \mathbf{z}$. Then we have

$$(1 - \alpha)z_j + \frac{\alpha}{m} \sum_{k=1}^{m} z_k \geq (1 - \alpha)z_j^* + \frac{\alpha}{m} \sum_{k=1}^{m} z_k^* \tag{51}$$

for every $j \in \{1, \ldots, m\}$.

Consequently,

$$\begin{aligned} w_j^i \cdot \tilde{z}_j = w_j^i \cdot \tilde{z}_j^* = 0, & \quad \text{if } w_j^i = 0, \\ w_j^i \cdot \tilde{z}_j \geq w_j^i \cdot \tilde{z}_j^*, & \quad \text{if } w_j^i > 0. \end{aligned} \tag{52}$$

Since $\exists j \in \{1, \ldots, m\} \setminus \{i\}$ such that $\tilde{z}_j^* > 0$, we can deduce

$$\max_{1 \leq j \leq m} \left\{ w_j^i \cdot \tilde{z}_j \right\} > \max_{1 \leq j \leq m} \left\{ w_j^i \cdot \tilde{z}_j^* \right\} \geq 0. \tag{53}$$

$\mathbf{z}$ can not be optimal for the boundary subproblem. Therefore, the optimal solution to the boundary subproblem must be cone-optimal. According to Corollary 2, the optimal solution is also properly Pareto-optimal. $\square$

## F.10   Proof of Theorem 6

*Proof.* Let $z_j^r = 0$ for $j = 1, \ldots, m$. Suppose that $\mathbf{z}$, rather than $\mathbf{z}^*$, is optimal for the subproblem with $\mathbf{w}^*$. Letting $w_j^* \tilde{z}_j^* = \beta$ for $j \in \{1, \ldots, m\} \setminus \{i\}$ where $\beta > 0$ is a constant, we have

$$\begin{aligned} w_j^* \cdot \tilde{z}_j^* = w_j^* \cdot \tilde{z}_j = 0, & \quad \text{if } w_j^* = 0, \\ \beta > w_j^* \cdot \tilde{z}_j \geq 0, & \quad \text{if } w_j^* > 0. \end{aligned} \tag{54}$$

Since $\mathbf{z}^*$ is properly Pareto-optimal according to Theorem 5, we have $w_j^* > 0, j \neq i$. We can deduce that $\tilde{z}_j^* > \tilde{z}_j \geq 0$ for $j \in \{1, \ldots, m\} \setminus \{i\}$. For the subproblem with $\mathbf{w}^i$ we obtain

$$\max_{1 \leq j \leq m} \left\{ w_j^i \cdot \tilde{z}_j^* \right\} > \max_{1 \leq j \leq m} \left\{ w_j^i \cdot \tilde{z}_j \right\}, \tag{55}$$

which implies $\mathbf{z}$, rather than $\mathbf{z}^*$, is the optimal objective vector to the boundary subproblem with $\mathbf{w}^i$. This is a contradiction. $\square$

