# OpenReview forum: "Boundary Decomposition for Nadir Objective Vector Estimation"
_NeurIPS.cc/2024/Conference — NeurIPS 2024 poster_

### Official Review · Reviewer_2guY · 2024-07-06

**Soundness:** 2
**Presentation:** 2
**Contribution:** 2
**Rating:** 6
**Confidence:** 4

**Summary:**

This paper proposes a nadir point estimation method for evolutionary multi-objective optimization, named BDNE. BDNE decomposes the MOP into boundary subproblems and uses a bilevel architecture to optimize them. Theoretical analysis and empirical results demonstrate the effectiveness of the proposed method.

**Strengths:**

This paper studies a problem: nadir point specification. A method based on decomposition is proposed, which is different from the past approaches. Some mathematical properties of the proposed decomposition method are demonstrated. An empirical study is conducted to highlight the effectiveness of the proposed method.

**Weaknesses:**

Generally, this paper is not well written, and the empirical study is not well organized.

## Presentation
I believe that interested readers can easily follow the authors' reasoning from the Abstract to Section 2, and the research background of this paper is also explained very clearly. However, starting from Section 3, it becomes very confusing. In Section 3.1, the authors introduce the boundary subproblem (Eq. 3) without providing any motivation for doing so. Although the authors title Section 2 as Motivation, I feel it actually serves as Related Work because this part only summarizes some previous work and does not touch on how the authors are thinking about this problem from a new perspective. After Eq. 3, the authors propose a new dominance relation and its three properties. I am very curious why the authors emphasize these three properties, as they do not seem to be used later in the paper. Subsequently, the authors present some theorems but do not sufficiently explain the relationship between these theorems and the new method proposed in this paper. In Section 3.2, the authors use a new boundary subproblem (Eq. 9) instead of Eq. 3 discussed in Section 3.1, which makes it unclear whether these mathematical properties in Section 3.1 are related to the proposed BDNE algorithm. Additionally, these properties only hold under very ideal assumptions (L128-L130).

Moreover, I suggest that the authors use some illustrations when introducing these new concepts, such as the boundary subproblem and cone-domination, to better help readers understand these concepts. Additionally, it would be worthwhile to explain the motivation behind them. Another area for improvement is the tables in the paper. The authors have recorded the experimental data of the baseline and BDNE in separate tables (Table 3 and Table 4), which makes it difficult for readers to compare the numbers. The authors could merge these two tables into one, making the data comparison more intuitive. Tables 7-9 have the same issue.

## Experiment
The empirical study of this paper does not adequately demonstrate the significance of BDNE in MOO. The experiments only show that BDNE can determine the nadir point more accurately than the baseline. However, I believe that many readers are more interested in the significance of an accurate nadir point for MOAs, such as whether it can improve the algorithm's convergence ability or enhance the diversity of the solution set. Currently, most MOAs do not have an independent mechanism to determine the nadir point, so now there might not have been a consensus in the MOO field on the necessity and significance of such an independent algorithm that is designed just to get the nadir point. Therefore, this paper is expected to demonstrate this clearly. For instance, the authors mention that the nadir point can be used for normalization, but normalization may not require a very accurate estimate of the nadir point. A rough estimation usually works as well. Many algorithms do not even have a normalization process. In practical applications, the scales of different objectives can be adjusted manually. In summary, I believe that the contribution of this paper to the field of MOO is limited and the authors did not adequately demonstrate its practical value.

Here are some minor remarks for the experiment part:
* What criteria were used to select these test problems? These test problems seem to be chosen arbitrarily or selected based on benchmark results after the fact.
* In the experiments, 180,000 FEs were used just to find the nadir point. Isn't this too many? Current SOTA MOAs generally require less than 50,000 FEs to obtain a well-converged and uniformly distributed solution set for these test problems.
* Why do some test problems have 3, 5, and 8 objectives, while some only have 3 objectives, e.g., both MaF2 and DTLZ5 can be configured with any number of objective functions, but they only have 3 objectives in the experiment.

**Questions:**

Please refer to *Weaknesses*.

**Limitations:**

No concerns.

---

> ### Author Rebuttal · Authors · 2024-08-07
>
> Thank you very much for your time and effort in reviewing our work. We are glad to know that you find our method is novel and has theoretical guarantees.
>
> We address your concerns as follows.
>
> ## W1. Presentation.
> **Transition from Section 2 to Section 3.**
> Thank you for raising this concern. It is better to choose ``Related Works'' as the title of Section 2. We will also include transitional statements at the beginning of Section 3.
>
> **Role of properties.**
> The three properties imply that cone domination defines a strict partial order. These properties are not utilized subsequently. We apologize for the implicit statement and will simplify the description based on your response.
>
> **Relationship between theorems and the new method.**
> We believe the relationship is clearly stated. Each theorem is associated with a corresponding explanation or utilized in a specific context. We may improve clarity by providing a summary.
>
> Theorems 1 and 2 are primary theoretical foundations for employing BDNE to determine the nadir objective vector. The user-friendly parameter $\mu$ in BDNE is proposed based on Theorem 3. Theorem 5 is instrumental in some strategies of BDNE (e.g., the coping strategy for flat fitness). Theorem 4 extends Theorem 1. While it is not directly related to BDNE, it is crucial for proving Theorem 5.
>
> **Illustrations of new concepts.**
> Thank you for the suggestion. We will show the level surfaces of boundary subproblems and their optimal solutions on an example $PF$ (see Figure 1 of the one-page PDF). Cone domination and proper Pareto optimality are not new concepts. We have cited relevant studies in our paper.
>
> **Motivation behind mentioning concepts.**
> Cone domination has a strong connection with the boundary subproblem as reported in Theorem 2, which is the motivation for mentioning this concept. Proper Pareto optimality is introduced to replace the obscure parameter (i.e., $\alpha$) with an easy-to-understand one (i.e., $\mu$). We will improve the clarity in the revised paper.
>
> **Tables.**
> Thank you for the suggestion. We will present the results in one table. An example is Table 1 of the one-page PDF.
>
> ## W2. Significance of the accurate nadir objective vector.
> We agree with you that normalization may not require an exact nadir objective vector. However, an accurate nadir objective vector is important in many scenarios and many exact methods are proposed in [1,2]. We list some consequences of inaccurate estimation of the nadir objective vector as follows:
>
> * Inaccurate nadir objective vectors **degrade** the performance of exact algorithms [3], evolutionary algorithms [4], and multi-objective learning [5].
> * An accurate nadir objective vector is often the **assumption** of interactive algorithms [6,7]. Inaccurate nadir objective vectors may cause biased decisions.
>
> Existing exact methods have limitations in applicability (e.g., continuous problems). Heuristic methods only perform well on manually designed benchmark problems since they often have simple feasible objective regions [4,8,9]. Real-world problems usually have irregular feasible objective regions, making their nadir objective vectors difficult to estimate [10,11]. Therefore, we propose BDNE, which is **the first method with theoretical guarantees and general applicability** in multi-objective optimization. Our method can easily adopt various solvers for different tasks and achieve a trade-off between runtime and accuracy.
>
> ## Q1. Eq. (3) versus Eq. (9).
> The mathematical properties of Eq. (3) remain valid for Eq. (9). The solutions associated with the critical points are invariant after the normalization (i.e., scaling and translation), which is straightforward. We further explain the reasonability behind this conversion in Lines 185-188.
>
> ## Q2. Ideal assumptions in Lines 128-130.
> The statements in Lines 128-130 refer to an ideal method, **not** ideal assumptions. This ideal method aims to demonstrate the preliminary feasibility of boundary decomposition in identifying the nadir objective vector. It motivates us to propose the BLOP, which is a practical optimization task.
>
> ## Q3. Criteria used to select test problems.
> Please refer to Common Concern 3.
>
> ## Q4. Too many function evaluations.
> Our method can achieve a trade-off between runtime and accuracy. We aim to ensure the algorithm converges sufficiently, validating our theoretical claims empirically. Consequently, we allocate a large number of function evaluations for each algorithm.
>
> ## Q5. Results on scalable problems with 3, 5, and 8 objectives.
> Our theoretical results hold for any number of objectives. You can refer to Common Concern 3 for an explanation and the supplementary experiments.
>
> ## References
>
> [1] Computing the nadir point for multiobjective discrete optimization problems. Journal of Global Optimization, 2015.
>
> [2] On nadir points of multiobjective integer programming problems. Journal of Global Optimization, 2017.
>
> [3] New $\epsilon$-constraint methods for multi-objective integer linear programming: A Pareto front representation approach. EJOR, 2023.
>
> [4] A new two-stage evolutionary algorithm for many-objective optimization. TEVC, 2019.
>
> [5] Hypervolume maximization: A geometric view of Pareto set learning. NeurIPS, 2023.
>
> [6] Multiobjective optimization: Interactive and evolutionary approaches. 2008.
>
> [7] A mini-review on preference modeling and articulation in multi-objective optimization: current status and challenges. Complex \& Intelligent Systems, 2017.
>
> [8] Regular Pareto front shape is not realistic. CEC, 2019.
>
> [9] A survey of normalization methods in multiobjective evolutionary algorithms. TEVC, 2021.
>
> [10] An easy-to-use real-world multi-objective optimization problem suite. Applied Soft Computing, 2020.
>
> [11] A benchmark-suite of real-world constrained multi-objective optimization problems and some baseline results. Swarm and Evolutionary Computation, 2021.

---

> ### Comment · Reviewer_2guY · 2024-08-09
>
> Thank you for your response. My primary concern is that the authors have not adequately addressed the practical significance of an exact nadir point. I would like to emphasize that most existing multi-objective optimization algorithms estimate the nadir point based on the current solution set. While I acknowledge that such estimates are not accurate, they do not constitute a bottleneck for these algorithms. I recognize that the exact calculation of the nadir point is an interesting problem, but I believe that the contribution of this paper to the field of multi-objective optimization is very limited. The empirical study in this paper only demonstrates that the proposed method could accurately calculate the nadir point, but not how the proposed method could improve the performance of a multi-objective optimization algorithm.

---

> ### Author Response · Authors · 2024-08-10
> **Thank you very much for your comment [1/2].**
>
> Thanks for your further comments. Below is our point-to-point response.
>
> ``Practical significance of an exact nadir point.``
>
> Firstly, we must clarify that our method is not just an exact algorithm, and its accuracy and runtime can be balanced via a controllable parameter. Secondly, we do not intend to claim that the accurate nadir objective vector is necessary for **all** multi-objective optimization scenarios. What we emphasize is that we should not ignore the cases where the accurate nadir objective vector is essential. There have already been a lot of works devoted to accurate nadir objective vector estimation [1-13]. We provide the following three examples:
>
> * The nadir objective vector frequently serves as the reference point of scalarization methods. Some applications require specifying an accurate nadir objective vector beforehand [10,11]. An underestimated nadir objective vector makes some Pareto-optimal solutions unattainable, whereas an overestimated one results in poor uniformity of obtained objective vectors.
> * The nadir objective vector determines the search space for some exact methods, such as AUGM-2. Exact methods without an accurate nadir objective vector may skip some Pareto-optimal solutions or exhibit longer runtimes. A recent experimental analysis can be found in [12].
> * Many interactive methods, such as the NIMBUS method, require accurate nadir values to present the promising region of the objective space for decision-makers [13]. If inaccurate nadir values are given, more interactions are conducted or a biased decision is made.
>
> [1] Ehrgott et al. Computation of ideal and nadir values and implications for their use in MCDM methods. EJOR, 2003.
>
> [2] Klamroth et al. Integrating Approximation and Interactive Decision Making in Multicriteria Optimization. OR, 2007.
>
> [3] Dhaenens et al. K-PPM A new exact method to solve multi-objective combinatorial optimization problems. EJOR, 2010.
>
> [4] Florios et al. Generation of the exact Pareto set in Multi-Objective Traveling Salesman and Set Covering Problems. Applied Mathematics and Computation, 2014.
>
> [5] Kirlik et al. Computing the nadir point for multiobjective discrete optimization problems. Journal of Global Optimization, 2015.
>
> [6] Köksalan et al. Finding nadir points in multi-objective integer programs. Journal of Global Optimization, 2015.
>
> [7] Boland et al. A new method for optimizing a linear function over the efficient set of a multiobjective integer program. EJOR, 2017.
>
> [8] Özpeynirci et al. On nadir points of multiobjective integer programming problems. Journal of Global Optimization, 2017.
>
> [9] Altamiranda et al. A New Exact Algorithm to Optimize a Linear Function over the Set of Efficient Solutions for Biobjective Mixed Integer Linear Programs. INFORMS Journal on Computing, 2020.
>
> [10] Zhang et al. Hypervolume maximization: A geometric view of Pareto set learning. NeurIPS, 2023.
>
> [11] Mena et al. Multi-objective two-stage stochastic unit commitment model for wind-integrated power systems: A compromise programming approach. International Journal of Electrical Power & Energy Systems, 2023.
>
> [12] Mesquita-Cunha et al. New $\epsilon$-constraint methods for multi-objective integer linear programming: A Pareto front representation approach. EJOR, 2023.
>
> [13] Branke et al. Multiobjective optimization: Interactive and evolutionary approaches. 2008.
>
>
> ``Most multi-objective optimization algorithms estimate the nadir point based on the current solution set.``
>
> We respectfully disagree with this statement. Many works show that only multi-objective evolutionary algorithms often use the current solution set to estimate the nadir objective vector [1]. Other widely adopted nadir objective vector estimation methods include the pay-off table method [2], extreme-point-based methods [3], subset-selection-based methods [4], and exact methods [5].
>
> [1] He et al. A survey of normalization methods in multiobjective evolutionary algorithms. TEVC, 2021.
>
> [2] Reeves et al. Minimum values over the efficient set in multiple objective decision making. EJOR, 1988.
>
> [3] Deb et al. An evolutionary many-objective optimization algorithm using reference-point-based nondominated sorting approach, part I: Solving problems with box constraints. TEVC, 2014.
>
> [4] Mallipeddi et al. A twin-archive guided decomposition based multi/many-objective evolutionary algorithm. Swarm and Evolutionary Computation, 2022.
>
> [5] Özpeynirci et al. On nadir points of multiobjective integer programming problems. Journal of Global Optimization, 2017.

---

> ### Author Response · Authors · 2024-08-10
> **Thank you very much for your comment [2/2].**
>
> ``Inaccurate estimates do not constitute a bottleneck for multi-objective optimization algorithms.``
>
> We also respectfully disagree with this statement. Experimental results on existing benchmarks cannot reflect the algorithm's real capability, as these benchmarks are manually designed with special features and can only be used to evaluate the algorithm's specific properties [1]. We provide the following experiment to demonstrate that inaccurate nadir objective vector estimation can have a significant impact on algorithm performance. We consider MOEA/D and NSGA-III. We replace their nadir objective vector estimation methods with ours and denote the variant as "V1". The mean and standard deviation of the hypervolume metric values are reported in the table. We can find that both MOEA/D-V1 and NSGA-III-V1 significantly outperform their original versions, respectively. Additionally, MOEA/D-V1 exhibits clearly smaller standard deviations compared with its original versions (e.g., 0.01 versus 0.003), indicating more stable performance. NSGA-III-V1 also yields stable performance.
>
> |Problem|$m$|MOEA/D|MOEA/D-V1|
> |--|--|--|--|
> |TN1|3|0.3891±0.01062(2)-|0.3986±0.002891(1)|
> |TN2|3|0.2166±0.01537(2)-|0.2271±0.002339(1)|
> |TN3|3|0.389±0.01352(2)-|0.3987±0.004753(1)|
> |TN4|3|0.2135±0.02198(2)-|0.2269±0.002335(1)|
>
> |Problem|$m$|NSGA-III|NSGA-III-V1|
> |--|--|--|--|
> |TN1|3|0.3361±0.02089(2)-|0.3769±0.003307(1)|
> |TN2|3|0.2114±0.01662(1)=|0.2071±0.002607(2)|
> |TN3|3|0.3383±0.02713(2)-|0.3782±0.003549(1)|
> |TN4|3|0.1442±0.02976(2)-|0.2096±0.003082(1)|
>
> We hope the above experiment can help you better understand the significance of the accurate nadir objective vector.
>
> [1] Ishibuchi et al. Performance of Decomposition-Based Many-Objective Algorithms Strongly Depends on Pareto Front Shapes. TEVC, 2017.
>
> ``Limited contribution to multi-objective optimization.``
>
> We cannot agree with you on this point as well. Nadir point estimation has always been a fundamental yet important research direction in multi-objective optimization, and many papers have been published on this topic. We search on Google Scholar using *("nadir point" OR "nadir objective vector") AND ("estimate" OR "find" OR "compute")* and return over 8,000 results. Compared with existing methods, our method is the first work with theoretical guarantees and general applicability. We would like to emphasize that our method is not only able to accurately estimate the nadir objective vector but also can trade off the accuracy for faster runtime. Our method can cope with scenarios that require the accurate nadir objective vector while ensuring broad applicability to cases that do not need an accurate nadir objective vector.
>
> ``How the proposed method could improve the performance of an algorithm?``
>
> We further improve NSGA-III-V1 by focusing the search within the promising region defined by the nadir objective vector. The statistical results of the hypervolume metric are presented in the following table. The improved version of NSGA-III-V1 is termed NSGA-III-V2. According to the results, our method shows great potential to improve the performance of an algorithm. Moreover, we find that NSGA-III-V2 retains more solutions within the promising region compared with NSGA-III-V1 (30 versus 39 on TN1, 25 versus 32 on TN2, 32 versus 38 on TN3, and 25 versus 33 on TN4). This explains the performance improvement and demonstrates the effectiveness of the scheme.
>
> |Problem|$m$|NSGA-III|NSGA-III-V1|NSGA-III-V2|
> |--|--|--|--|--|
> |TN1|3|0.3361±0.02089(3)-|0.3769±0.003307(2)-|0.3849±0.003166(1)|
> |TN2|3|0.2114±0.01662(2)-|0.2071±0.002607(3)-|0.2125±0.002711(1)|
> |TN3|3|0.3383±0.02713(3)-|0.3782±0.003549(2)-|0.3821±0.003803(1)|
> |TN4|3|0.1442±0.02976(3)-|0.2096±0.003082(2)-|0.213±0.00347(1)|

---

> > ### Comment · Reviewer_2guY · 2024-08-10
> >
> > Thank you for your response. This new experiment is very meaningful. Can you report the FE budget for this experiment?
> > Do MOEA/D and MOEA/D-V1 use the same budget (including those used in the nadir point estimation)?

---

> > > ### Author Response · Authors · 2024-08-10
> > >
> > > Thank you very much for your prompt reply. The number of function evaluations is the same as in our original manuscript, i.e., 180,000. We confirm that both MOEA/D and MOEA/D-V1 use the same FE budget. MOEA/D-V1 consumes 12 FEs for nadir objective vector estimation in each iteration and uses fewer iterations (about 230 less than MOEA/D).

---

> > > ### Author Response · Authors · 2024-08-11
> > >
> > > We notice that your rating remains unchanged. Could you please let us know if your previous concerns have been addressed?
> > >
> > > If you have any new concerns, please let us know as well.

---

> > > > ### Comment · Reviewer_2guY · 2024-08-11
> > > >
> > > > Thank you for your response. Some of my concerns have been adequately addressed. However, the authors mentioned that there are around 8,000 papers focusing on nadir point estimation. However, there are only two baselines presented in this paper, and they were 7 years ago. I think the empirical study in this paper can be further improved, and the current evaluation is not very comprehensive. I am still worried about whether the method proposed in this paper is more effective than existing ones.

---

> > > > > ### Author Response · Authors · 2024-08-12
> > > > >
> > > > > Thank you very much for your feedback, and we are pleased to hear that some of your concerns have been adequately addressed. We address your remaining concerns as follows.
> > > > >
> > > > > In our original manuscript, we mainly use theoretical analysis and illustrative examples to demonstrate the advantage of our method over the existing heuristic ones. We believe this is sufficient to validate the shortcomings of these current methods. Therefore, we only use two representative baselines in our empirical study. To more comprehensively illustrate the effectiveness of our method, we present the experimental results of the other heuristic methods in the following table. We can find that our method still outperforms the other methods.
> > > > >
> > > > >
> > > > > | Problem      | $m$ | 2017 [1]         | 2018 [2]          | 2019 [3]            | 2019 [4]             | 2021 [5]          | 2022 [6]                | 2022 [7]            | 2023 [8]          | BDNE                   |
> > > > > |--------------|-----|------------------|-------------------|---------------------|----------------------|-------------------|-------------------------|---------------------|-------------------|------------------------|
> > > > > | TN1          | 3   | 2.232±1.36(9)-   | 0.8477±0.2477(7)- | 1.044±0.2409(8)-    | 0.2026±1.58e-07(5)-  | 0.3735±0.3905(6)- | 1.708e-15±3.328e-15(2)= | 0.09419±0.1188(3)-  | 0.1704±0.2911(4)- | 3.144e-16±6.649e-16(1) |
> > > > > | TN2          | 3   | 1.456±0.9491(9)- | 0.5103±0.2975(7)- | 1.054±8.842e-15(8)- | 0.2231±3.543e-07(3)- | 0.437±0.2457(6)-  | 0.3333±0(5)-            | 0.1506±0.28(2)-     | 0.266±0.211(4)-   | 0.0001722±0.0005349(1) |
> > > > > | TN3          | 3   | 2.151±1.429(9)-  | 0.7755±0.2774(7)- | 1±0(8)-             | 0.2628±4.023e-08(5)- | 0.2744±0.2297(6)- | 6.388e-16±9.846e-16(2)= | 0.04554±0.03419(3)- | 0.1234±0.1045(4)- | 2.388e-16±2.32e-16(1)  |
> > > > > | TN4          | 3   | 1.348±0.8802(9)- | 0.4879±0.3249(7)- | 1.054±1.748e-15(8)- | 0.3367±4.039e-08(5)- | 0.4094±0.1397(6)- | 0.3333±0(4)-            | 0.1554±0.265(2)-    | 0.2631±0.1829(3)- | 4.924e-05±5.587e-05(1) |
> > > > > | Average rank |     | 9(9)             | 7(7)              | 8(8)                | 4.5(5)               | 6(6)              | 3.25(3)                 | 2.5(2)              | 3.75(4)           | 1(1)                   |
> > > > >
> > > > >
> > > > > In Common Concern 2, we also present experimental results demonstrating the promising performance of our method in comparison to exact methods. We will supplement these experiments in the revised paper and release our code upon publication. We hope the above discussion can address your remaining concerns. Please let us know if you have any further concerns.
> > > > >
> > > > > [1] Nadir point estimation for many-objective optimization problems based on emphasized critical regions. Soft Computing, 2017.
> > > > >
> > > > > [2] A constrained decomposition approach with grids for evolutionary multiobjective optimization. TEVC, 2018.
> > > > >
> > > > > [3] Distance-based subset selection for benchmarking in evolutionary multi/many-objective optimization. TEVC, 2019.
> > > > >
> > > > > [4] IGD indicator-based evolutionary algorithm for many-objective optimization problems. TEVC, 2019.
> > > > >
> > > > > [5] A many-objective evolutionary algorithm based on a two-round selection strategy. TCYB, 2021.
> > > > >
> > > > > [6] Multiobjective Optimization-Aided Decision-Making System for Large-Scale Manufacturing Planning. TCYB, 2022.
> > > > >
> > > > > [7] A twin-archive guided decomposition based multi/many-objective evolutionary algorithm. Swarm and Evolutionary Computation, 2022.
> > > > >
> > > > > [8] A many-objective evolutionary algorithm with adaptive convergence calculation. Applied Intelligence, 2023.

---

> ### Comment · Reviewer_2guY · 2024-08-12
>
> Thank you for your response. Now most of my major concerns have been satisfactorily addressed, so I increased my rating. I know it is very difficult to prepare the additional results during the rebuttal period, so I do appreciate your efforts. However, I strongly suggest the authors incorporate a more comprehensive and solid empirical study in the camera-ready version (if accepted) including a comparison with other strong baselines on more test problems and an ablation study to explore how BDNE helps improve the performance of MOAs in practical scenarios. I think a wide range of audiences may be interested in these results.

---

> > ### Author Response · Authors · 2024-08-12
> >
> > We sincerely thank you for raising your score and are glad to know your major concerns have been satisfactorily addressed. We will include a more comprehensive empirical comparison in our final version. Thank you very much for your time and valuable suggestions, which will greatly help us improve our manuscript.

---

### Official Review · Reviewer_wp71 · 2024-07-08

**Soundness:** 3
**Presentation:** 3
**Contribution:** 3
**Rating:** 7
**Confidence:** 5

**Summary:**

The authors model the task of computing the nadir objective vector as several bilevel optimization problems.  A corresponding algorithm named BDNE is designed to estimate the nadir objective vector for black-box multi-objective optimization problems.  BDNE scalarizes a multiobjective optimization problem into a set of boundary subproblems.  By utilizing bilevel optimization on the boundary subproblems, the nadir objective vector is identified.  To demonstrate the performance, the new approach is applied to some benchmark problems and real-world problems.

**Strengths:**

The authors introduce bilevel optimization problems, theoretically ensuring that the nadir objective vector is solved.  Although the corresponding algorithm is an approximation algorithm, experimental results show that it can significantly outperform existing heuristic methods.  Moreover, the article is reasonably well organized.  The examples are detailed clearly, as is the new approach.

**Weaknesses:**

1. The nadir objective vector estimation is just part of the process of solving the original multi-objective optimization problem.  The development of BDNE may be not highly significant.
2. m boundary weight vectors remain unchanged throughout.  It may waste the computational resources.
3. Solving the original problem creates a new optimization problem, potentially increasing complexity.  And BDNE does require a longer runtime than the compared algorithms.
4. The illustration of setting mu is a bit preliminary.

**Questions:**

1. Why fix m boundary weight vectors as unit vectors?  Can they change after some iterations?
2. The setting of mu should have a further discussion.
3. What increases the computational cost of BDNE?  The computational complexity of BDNE should be analyzed.
4. What advantages does this method have compared to existing exact algorithms when applied to discrete problems?

**Limitations:**

The longer runtime of the proposed algorithm might also be a limitation that is not thoroughly discussed in the paper.

---

> ### Author Rebuttal · Authors · 2024-08-07
>
> Thank you very much for your time and effort in reviewing our work. We are glad to know that you find this work is well organized, the analysis of existing methods is detailed and clear, and our method is demonstrated theoretically and experimentally.
>
> we address your concerns as follows.
>
> ## W1. Significance of BDNE.
> The nadir objective vector is an essential component of several fundamental operations (e.g., normalization, interaction, and guided search) in multi-objective optimization. In many engineering problems, the nadir objective vector has already been utilized to facilitate optimization [1,2,3].
>
> Despite the introduction of many exact and heuristic methods, a unified paradigm to estimate the nadir objective vector remains absent. Our method is **the first one with theoretical guarantees and general applicability** in multi-objective optimization. Our method can be applied to various scenarios by employing appropriate solvers. Furthermore, the trade-off can be easily determined by decision-makers via setting a parameter.
>
> ## W2/Q1. Unchanged boundary subproblems waste computational resources. Can they change after some iterations?
> These unchanged boundary subproblems use the unit vectors as their weight vectors. They motivate the search for the ideal objective vector, which serves as $\mathbf{z}^{r1}$. Besides, it enhances population diversity when the adjustable boundary subproblems become similar. In Table 1 of the one-page PDF, BDNE-V2 denotes BDNE removing boundary subproblems with unit vectors. We can find that BDNE outperforms BDNE-V2 across most instances. The effectiveness of these subproblems is demonstrated empirically.
>
> Changing these subproblems after some iterations is an interesting idea. Further investigations will be conducted to explore this.
>
> ## W3/Q3. Longer runtime.
> The runtime for BDNE strongly depends on the implementation. The main loop of our implementation has a time complexity of $O\left(N^2\log(N)\right)$ where $N$ is the population size. The specific analysis is provided as follows. The reproduction procedure has a time complexity of $O(N)$. The calculation of $\mathbf{r}$ governs the complexity of the selection procedure, which has $O\left(N^2\log(N)\right)$. The adjustment of boundary weight vectors with $O\left(N\log\left(\frac{N}{m}\right)\right)$ is executed periodically. Therefore, **The selection procedure governs the overall complexity**. The main loop has the complexity $O\left(N^2\log(N)\right)$. We will include the computational complexity analysis in the revised version.
>
> We use an effective selection procedure stated in [4], which can facilitate the application of BDNE to various problems. We can significantly reduce the complexity by using a **simple selection procedure** or choosing an **alternative solver**, which deserves further investigation.
>
> ## W4/Q2. Criteria of setting $\mu$.
> $\mu>0$ is a user-defined parameter, indicating a bounded trade-off preferred by decision-makers. Once $\mu$ is determined, $\alpha$ in Eq. (3) can be calculated (see Lines 155-162). If the preferences of decision-makers are not available, $\mu$ can be set to a reasonably large value. In the experimental studies, we demonstrate that BDNE with $\mu=100$ can have an overall good performance. We will improve the clarity in the revised paper.
>
> ## Q4. BDNE versus exact methods on discrete problems.
> A comparison of our methods to existing exact methods is summarized in the following table. The exact method involves a more complicated optimization task, which includes more single-objective optimization problems or slack variables. Some exact methods require the variable to be discrete. Moreover, existing exact methods do not accommodate a user-defined trade-off.
>
> |    Method   | Number of single-objective optimization problems | Number of slack variables | Discrete variable | User-defined trade-off |
> |:-----------:|:------------------------------------------------:|:-------------------------:|:-----------------:|:---------------------------------------:|
> |    KS [5]    |                         2                        |            Null           |    Unnecessary    |               Incompatible              |
> |    KL [6]    |                         2                        |         Increasing        |     Necessary     |               Incompatible              |
> |   FD&IS [7]  |                         2                        |         Increasing        |     Necessary     |               Incompatible              |
> | BDNE (ours) |                         1                        |            Null           |    Unnecessary    |                Compatible               |
>
> We choose an exact solver for BDNE to validate its effectiveness empirically. BDNE has a leading performance in terms of runtime (please refer to Common Concern 2).
>
> ## References
> [1] Weerasuriy et al. Performance evaluation of population-based metaheuristic algorithms and decision-making for multi-objective optimization of building design. Building and Environment, 2021.
>
> [2] Mena et al. Multi-objective two-stage stochastic unit commitment model for wind-integrated power systems: A compromise programming approach. International Journal of Electrical Power \& Energy Systems, 2023.
>
> [3] Ekhtiari et al. Optimizing the dam site selection problem considering sustainability indicators and uncertainty: An integrated decision-making approach. Journal of Cleaner Production, 2023.
>
> [4] Zheng et al. A generalized scalarization method for evolutionary multi-objective optimization. AAAI, 2023.
>
> [5] Kirlik et al. Computing the nadir point for multiobjective discrete optimization problems. Journal of Global Optimization, 2015.
>
> [6] Köksalan et al. Finding nadir points in multi-objective integer programs. Journal of Global Optimization, 2015.
>
> [7] Özpeynirci et al. On nadir points of multiobjective integer programming problems. Journal of Global Optimization, 2017.

---

> > ### Comment · Reviewer_wp71 · 2024-08-10
> >
> > Thank you for your reply and additional experiments with the discrete problem, which make me more explicit about the significance of the proposed BNDE. However, I am still confused about the "trade-off" in W1 and W4/Q2. If the authors could provide more explanations or illustrations, I would consider raising my rating.

---

> > > ### Author Response · Authors · 2024-08-11
> > >
> > > Thanks for your further comments. We apologize for the unclear expression. The two trade-offs are different.
> > >
> > > ``Trade-off in W1.``
> > >
> > > This refers to the trade-off between the accuracy of the estimated nadir objective vector and the runtime of BDNE. This trade-off can be controlled by the parameter $\tau_u$. A larger value of $\tau_u$ results in higher accuracy, while a smaller value of $\tau_u$ leads to a shorter runtime. We provide the following experiments to show the effects of $\tau_u$. The first and second tables present statistical results of errors and runtimes, respectively. We can find that the runtime is getting shorter as the value of $\tau_u$ decreases; the error is getting smaller (or the accuracy is getting higher) as the value of $\tau_u$ increases.
> > >
> > > |Problem|$m$|2|6|10|14|18|
> > > |--------------|-----|--------------------|----------------------|------------------------|-----------------------|------------------------|
> > > |TN4|3|0.01654±0.01637(5)|0.001661±0.001799(4)|0.0001939±0.0002971(3)|0.000134±0.0004145(2)|5.646e-05±0.0002382(1)|
> > > ||5|0.08632±0.05066(5)|0.01003±0.01915(4)|0.009246±0.02555(2)|0.003051±0.007012(1)|0.009329±0.02633(3)|
> > > ||8|0.08955±0.09165(5)|0.03703±0.05067(3)|0.04032±0.08866(4)|0.03051±0.04605(2)|0.0255±0.02799(1)|
> > > |Average rank||5(5)|3.6667(4)|3(3)|1.6667(1)|1.6667(1)|
> > >
> > > |Problem|$m$|2|6|10|14|18|
> > > |--------------|-----|-----------------|-----------------|-----------------|----------------|----------------|
> > > |TN4|3|4.551±0.1162(1)|12.52±0.3078(2)|20.04±0.4353(3)|27.44±0.488(4)|34.89±0.645(5)|
> > > ||5|8.587±0.2431(1)|23.2±0.3991(2)|37.45±0.5952(3)|51.6±0.8106(4)|65.96±1.079(5)|
> > > ||8|16.22±0.2524(1)|44.8±0.6788(2)|72.78±1.144(3)|100.9±1.64(4)|129.1±1.898(5)|
> > > |Average rank||1(1)|2(2)|3(3)|4(4)|5(5)|
> > >
> > >
> > > ``Trade-off in W4/Q2.``
> > >
> > > This implies the trade-off between objectives. This trade-off is the parameter $\mu$, determined by the decision-maker. Specifically, $\mu$ is the amount of increment in the value of one objective function that the decision-maker is willing to tolerate **in exchange for a one-unit decrement in another objective function** (in the context of the minimization problem). Each Pareto-optimal solution corresponds to a particular value of $M$ (see Definition 10 in our original manuscript). If $M \leq \mu$, the Pareto-optimal solution aligns with the decision-maker’s preference. Otherwise, the Pareto-optimal solution should be discarded. We refer to the region where the objective vectors dominate the estimated nadir objective vector by BDNE as the promising region. Solutions outside this promising region can be ignored. Without the preference of $\mu$, $\mu$ is set to a reasonably large value, and BDNE finds the exact nadir objective vector. When this preference is available, the estimated nadir objective vector by BDNE may further narrow down the promising region.
> > >
> > > we conduct experiments to show the effects of $\mu$. The estimation errors are statistically shown in the following table. We can find that a smaller value makes a larger error. We also examine the position of the estimated nadir objective vector. The estimated one is dominated by the exact one on every instance. These findings indicate that a smaller value of $\mu$ can yield a more refined promising region.
> > >
> > > |Problem|$m$|1|5|10|50|100|
> > > |---------|---|---------------------|---------------------|---------------------|----------------------|-----------------------|
> > > |mDTLZ2|3|0.5211±0.0006548(5)|0.1548±0.001253(4)|0.0793±0.002107(3)|0.01325±0.001883(2)|0.006579±0.001046(1)|
> > > ||5|0.3927±0.0108(5)|0.08951±0.008372(4)|0.0471±0.003567(3)|0.01099±0.00667(2)|0.004697±0.0004873(1)|
> > > ||8|0.2884±0.01397(5)|0.06433±0.004405(4)|0.03188±0.002174(3)|0.005223±0.001297(2)|0.003585±0.004993(1)|

---

> > > > ### Comment · Reviewer_wp71 · 2024-08-12
> > > >
> > > > I thank the authors for their effective responses, and I have increased my score to 7 and will vote for acceptance. I appreciate the added experiments and the discussion, which are comprehensive and useful.
> > > > I suggest the authors add these discussions to the final version. Meanwhile, a more detailed summary of the advantages or weaknesses of the proposed method compared to existing ones would be beneficial to underscore this paper's significance and positioning.

---

> > > > > ### Author Response · Authors · 2024-08-12
> > > > >
> > > > > Thank you very much for raising your score and appreciating our previous response.
> > > > >
> > > > > We sincerely appreciate your valuable suggestions. We will add these discussions and revise our manuscript accordingly.

---

### Official Review · Reviewer_NNkD · 2024-07-08

**Soundness:** 3
**Presentation:** 4
**Contribution:** 4
**Rating:** 7
**Confidence:** 4

**Summary:**

This paper looks at the problem of finding the nadir objective vector in multi-objective optimization problems, which is important for both optimization and decision-making. This work first analyzes the drawbacks of existing techniques, and then proposes a new method for nadir objective vector estimation using bilevel optimization and a multi-objective evolutionary approach. Experimental results show that the proposed method can achieve good performance in several test examples.

**Strengths:**

This paper is very well written. As far as I can tell, the problem discussed in the paper is important for multi-objective optimization.

The analysis of existing methods is convincing. Particularly, the paper gives a clear example of poor estimation of the vector by existing heuristic techniques. The proposed method has theoretical guarantees and can be applied to a wide range of problems. The effectiveness of this method is also verified experimentally.

**Weaknesses:**

I do not find any obvious weakness, but I have the following questions:

1. Two different evolutionary algorithms are used to solve the upper- and lower-level problems respectively. Why is CMA-ES used for the optimization of upper-level problems but not for lower-level problems?

2. How should one select a value for the parameter mu in BDNE?

3. Suppose I select a relatively small value of mu and obtain a point consisting of the optimal values of bilevel optimization problems. Does this mean that in the area where the objective vectors do not dominate the point, there are no Pareto optimal objective vectors with a minimum M value less than mu?

3. I notice that the error on CRE5-3-1 is large (about 20%). Please explain this.

4. Why are algorithms only tested on continuous problems?

**Questions:**

Refer to the above comments.

**Limitations:**

The discussion in the conclusion section is comprehensive enough.

---

> ### Author Rebuttal · Authors · 2024-08-07
>
> Thank you very much for your time and effort in reviewing our work. We are glad to know that you find this work is important and well-written, the analysis of existing methods is convincing, and our method has wide applicability.
>
> we address your concerns as follows.
>
> ## Q1. Why is CMA-ES used for the optimization of upper-level problems but not for lower-level problems?
> We utilize CMA-ES as the optimizer for ULOP for two reasons. First, CMA-ES is a state-of-the-art algorithm for single-objective black-box optimization. Second, optimal solutions to the LLOPs are not always available; instead, approximate solutions are often obtained. Consequently, the function values of the approximate solutions may exhibit noise. CMA-ES is suitable for this task as it demonstrates strong robustness in optimizing noisy functions [1].
>
> CMA-ES can be used to solve the LLOP since the LLOP is a single-objective optimization problem. We utilize an MOEA to cooperatively solve the LLOPs. This is because the superiority of MOEAs is demonstrated empirically and theoretically in solving multi-objective black-box optimization problems [2,3].
>
> ## Q2. Criteria of setting $\mu$.
> $\mu>0$ is a user-defined parameter, indicating a bounded trade-off preferred by decision-makers. Once $\mu$ is determined, $\alpha$ in Eq. (3) can be calculated (see Lines 155-162). If the preferences of decision-makers are not available, $\mu$ can be set to a reasonably large value. In the experimental studies, we demonstrate that BDNE with $\mu=100$ can have an overall good performance.
>
> ## Q3. A small value of $\mu$: Suppose I select a relatively small value of $\mu$ and obtain a point consisting of the optimal values of bilevel optimization problems. Does this mean that in the area where the objective vectors do not dominate the point, there are no Pareto optimal objective vectors with a minimum $M$ value less than $\mu$?
> Yes. That is, the Pareto-optimal solution does not address the user-defined trade-off $\mu$ if it does not dominate the point constructed by the optimal values of BLOPs. The optimal solution of the $i$-th BLOP has the largest value of the $i$-th objective function among all the solutions addressing the trade-off. The conclusion can be deduced by Theorems 2 and 3. A satisfactory pseudo-nadir objective vector can be obtained if the decision-makers prefer a relatively small value of $\mu$. This preference-compatible property enables our method to have greater application potential. We will add the new claims and their proofs to the revised version.
>
> ## Q4. Large error on CRE5-3-1.
> Thank you for the careful review. The $PF$ of CRE5-3-1 is unknown and represented by the non-dominated objective vector set provided in [4]. The approximate $PF$ may be the reason for the large error since the results of DNPE and BDNE are similar.
>
> ## Q5. Why are algorithms only tested on continuous problems?
> We test BDNE on the discrete problem and choose an exact solver. Compared with exact methods, BDNE still has a superior performance in terms of runtime (please refer to Common Concern 2).
>
> ## References
> [1] Nikolaus Hansen. The CMA evolution strategy: A tutorial. arXiv preprint, 2016.
>
> [2] Zhang et al. MOEA/D: A multiobjective evolutionary algorithm based on decomposition. TEVC, 2007.
>
> [3] Dang et al. Crossover can guarantee exponential speed-ups in evolutionary multi-objective optimisation. AIJ, 2024.
>
> [4] Tanabe et al. An easy-to-use real-world multi-objective optimization problem suite. Applied Soft Computing, 2020.

---

> > ### Comment · Reviewer_NNkD · 2024-08-12
> >
> > Thank you for the rebuttal. My concerns have been resolved, so I will maintain the positive score.

---

> > > ### Author Response · Authors · 2024-08-12
> > >
> > > Thank you very much for your time and effort in reviewing our work, and we are glad to know your concerns have been resolved.

---

### Official Review · Reviewer_3RPX · 2024-07-08

**Soundness:** 3
**Presentation:** 3
**Contribution:** 3
**Rating:** 7
**Confidence:** 5

**Summary:**

This work proposes bilevel optimization problems to align their optimal values with the nadir point.  Some schemes are suggested to address potential flat fitness in upper-level optimization. An algorithm based on evolutionary computation is then proposed for black-box cases.

**Strengths:**

1. This work demonstrates both theoretically and experimentally that the nadir point can be obtained or approximated via newly proposed optimization problems.
2. Compared to existing optimization problems for finding the nadir point, the proposed problem is more concise and applicable to a broader range of scenarios.
3. Several schemes are used to cope with the issue of flat fitness in the upper-level problem.

**Weaknesses:**

1. The algorithm proposed in this paper is an approximation algorithm rather than an exact one.
2. The algorithm is general. Its performance may be unsatisfactory for some specific problems.
3. The lower-level optimization problem consists of a series of single-objective subproblems, limiting the applicability of some multi-objective optimization algorithms.

**Questions:**

1. Is there evidence of really complicated feasible objective spaces from industrial applications?
2. The algorithm for the lower-level optimization problem appears to be quite complicated.  Can we consider changing it to a different one?
3. Can evolutionary algorithms be used in the optimization problems proposed by Gokhan et al. (2015)?
4. How are the benchmarks chosen in the experiment?
5. What is the performance if coping strategies for flat fitness are eliminated?
6. I also wonder about the effects of removing the subproblems with unit vectors.
7. I recommend providing a more detailed analysis of the deficiencies of the heuristic methods.

**Limitations:**

This work mentions one limitation: the lack of fine-tuning to a specific problem.  I agree that this is an interesting and important topic for future research.

---

> ### Author Rebuttal · Authors · 2024-08-07
>
> Thank you very much for your time and effort in reviewing our work. We are glad to know that you find our method is concise, has broad applicability, and demonstrates its effectiveness theoretically and experimentally.
>
> we address your concerns as follows.
>
> ## W1/W2. BDNE is a general approximate algorithm.
> BDNE can **accurately** determine the nadir objective vector, as long as an exact solver is available. BDNE can achieve **satisfactory** performance for a specific task, as long as a suitable solver is utilized. This paper proposes a general framework applicable to various scenarios, such as multi-objective integer linear programming with available exact solvers, and multi-objective learning utilizing gradient information.
>
> To demonstrate that BDNE can be a promising exact algorithm, we apply BDNE to the discrete problem and choose an exact solver. We test BDNE on the multi-objective assignment problem (MOAP) and the multi-objective knapsack problem (MOKP). 3 instances are randomly generated for each problem. The following table shows the mean and standard deviation of error metric values. Our method is **accurate**, while the heuristic method misses the nadir objective vector. The superiority is also demonstrated, as shown in Common Concern 2.
>
> |Problem|$m$|$n$|ECR-NSGA-II|DNPE|BDNE|
> |:-------:|:---:|:------------:|:------------:|:--------------:|:----:|
> |MOAP|3|20$\times$20|0.261±0.103|0.392±0.0828|0|
> ||||0.202±0.064|0.419±0.055|0|
> ||||0.197±0.0945|0.582±0.0767|0|
> ||4|10$\times$10|0.147±0.0602|0.498±0.0405|0|
> ||||0.252±0.0917|1.16±4.52e-16|0|
> ||||0.142±0.0659|0.536±2.26e-16|0|
> |MOKP|3|200|0.218±0.0861|3.84±0.0284|0|
> ||||0.229±0.0804|4.62±0.0315|0|
> ||||0.208±0.084|5.48±0.0346|0|
> ||4|50|0.244±0.0957|3.43±0.0586|0|
> ||||0.27±0.0743|4.25±0.0587|0|
> ||||0.262±0.107|5.08±0.0709|0|
>
> ## W3. Applicability of other multi-objective optimization algorithms for lower-level optimization.
> We would like to clarify that the LLOPs constitute a set of single-objective optimization problems (SOPs) **rather than** an MOP. Therefore, not all multi-objective optimization algorithms are suitable for the lower-level optimization. Nevertheless, these SOPs can be solved sequentially or in parallel. We employ a decomposition-based multi-objective optimization algorithm to collaboratively address these SOPs, thereby enhancing optimization efficiency.
>
> ## Q1. Evidence of really complicated feasible objective region.
> Real-world problems usually have many constraints that can result in irregular feasible objective regions. A comprehensive review of these industrial cases is provided in [1,2].
>
> ## Q2. Can the complicated lower-level optimization algorithm be changed?
> Yes. We can change the selection strategy in this algorithm to reduce the complexity. This is because this algorithm employs an efficient selection strategy as described in [3], which governs the overall complexity. Moreover, the optimizer of LLOPs can be changed to any suitable one according to the task's properties. For example, use other decomposition-based evolutionary algorithm frameworks [4]. Local search (or gradient information, if available) can be employed to facilitate the convergence [5,6]. All single-objective optimizers such as GUROBI and BARON can also be applied to solve the LLOP, as the LLOP is a single-objective optimization problem.
>
> ## Q3. Can the method proposed in (Gokhan et al. 2015) adopt the evolutionary algorithm as the solver?
> This method relies heavily on an exact solver. This method is **time-consuming** and **unreliable** when the evolutionary algorithm is adopted. Specifically, its lower-level optimization problem, including two sequential single-objective optimization problems, exhibits high computational costs. Furthermore, uncertain optimality gaps arise when the evolutionary algorithm is adopted. As a result, the first single-objective optimization problem misleads the subsequent one, thereby presenting significant unreliability in the lower-level optimization process. The pay-off table, which is a key step, also cannot be accurately obtained.
>
> In contrast, our method does not require the payoff table, and the lower-level optimization problem is an independent single-objective optimization problem.
>
> ## Q4. Criteria used to select benchmarks.
> Please refer to Common Concern 3.
>
> ## Q5 \& Q6. Ablation studies.
> The experimental results are presented in Table 1 of the one-page PDF. BDNE-V1 denotes BDNE without the coping strategy for flat fitness. BDNE-V2 denotes BDNE removing boundary subproblems with unit vectors. The effectiveness of the two strategies is validated.
>
> ## Q7. More detailed analysis of deficiencies of the heuristic methods.
> We aim to point out that heuristic methods do not guarantee obtaining exact objective vectors. We construct an example $PF$ and then find all heuristic methods can not identify its nadir objective vector. We believe this example is clear and intuitive. Additional analysis is not necessary to support the above claim.
>
>
> ## References
> [1] Tanabe et al. An easy-to-use real-world multi-objective optimization problem suite. Applied Soft Computing, 2020.
>
> [2] Kumar et al. A benchmark-suite of real-world constrained multi-objective optimization problems and some baseline results. Swarm and Evolutionary Computation, 2021.
>
> [3] Zheng et al. A generalized scalarization method for evolutionary multi-objective optimization. AAAI, 2023.
>
> [4] Ke Li. A survey of decomposition-based evolutionary multi-objective optimization: Part I-past and future. arXiv preprint, 2024.
>
> [5] Lara et al. HCS: A new local search strategy for memetic multiobjective evolutionary algorithms. TEVC, 2010.
>
> [6] Lapucci et al. A memetic procedure for global multi-objective optimization. MPC, 2023.

---

> > ### Comment · Reviewer_3RPX · 2024-08-08
> >
> > Thanks for the response. I am glad to see that the proposed method can also work as an exact algorithm and achieve a shorter running time than other existing exact algorithms. The controllable trade-off between running time and accuracy is important and will greatly benefit the development of more advanced multi-objective optimization algorithms. My other concerns have been addressed as well. I will increase my score to 7 and support its acceptance.

---

> > > ### Author Response · Authors · 2024-08-08
> > >
> > > We sincerely appreciate your support and are delighted to hear that all your concerns have been addressed.

---

### Author Rebuttal · Authors · 2024-08-07

Dear AC and Reviewers,

We would like to thank you for the insightful comments and feedback on our paper. Overall, the reviewers agree that this work is important (NNkD) and well-written (NNkD, wp71), the analysis of existing methods is convincing (NNkD) /clear (wp71), the proposed method is concise (3RPX) /novel (2guY) and has broad applicability (3RPX, NNkD).

We address some common concerns raised by different reviewers in this response.

## C1. Significance (wp71, 2guY).
**Nadir objective vector.**
The nadir objective vector is a fundamental concept of multi-objective optimization, and it has been widely used in many multi-objective optimization methods (not only heuristic but also exact methods)'s operations, such as normalization, interaction, and guided search. There have been many studies demonstrating the importance of accurate estimation of the nadir objective vector [1,2,3] and its effectiveness in facilitating the optimization of real-world problems [4,5,6].

**BDNE.**
Heuristic methods do not have theoretical guarantees, while exact methods have limitations in applicability. Therefore, we propose BDNE, which is **the first method with theoretical guarantees and general applicability** in multi-objective optimization. Our method can easily adopt various solvers for different tasks and achieve a trade-off between runtime and accuracy. Furthermore, the trade-off can be easily determined by decision-makers via setting a parameter.


## C2. BDNE adopts an exact solver for discrete problems (3RPX, NNkD, wp71).
To demonstrate that BDNE can be competitive on other problems, we apply BDNE to the discrete problem and choose an exact solver. We test BDNE on the multi-objective assignment problem (MOAP) and the multi-objective knapsack problem (MOKP). 3 instances are randomly generated for each problem. The runtimes in seconds are shown in the following table. "TO" represents timeout. Compared with existing exact methods, BDNE still demonstrates superior performance in terms of runtime.

|Problem|$m$|$n$|KS [7]|KL [8]|FD&IS [9]|BDNE|
|:-------:|:---:|:------------:|:----:|:----:|:-----:|:-------:|
|MOAP|3|20$\times$20|340|4375|687|**222**|
||||405|1249|1471|**183**|
||||394|1214|915|**182**|
||4|10$\times$10|4279|575|TO|**212**|
||||8028|1189|TO|**273**|
||||4724|431|TO|**160**|
|MOKP|3|200|1703|270|826|**166**|
||||1976|77|238|**62**|
||||1500|131|283|**92**|
||4|50|1562|28|10211|**24**|
||||1106|53|14256|**36**|
||||412|21|3013|**15**|


## C3. Criteria used to select benchmarks (3RPX, 2guY).
Since some benchmark problems are designed too specially (e.g., triangle-like $PF$), they cannot reflect the difficulty for estimating the nadir objective vector of real-world problems. Therefore, the problems with complicated feasible objective regions are used in our experimental study. To comprehensively compare the algorithms and avoid showing similar results, we select the problems with different kinds of feasible objective regions from several test suites. Specifically, we employ 4 new and 6 existing test problems (28 instances in total), including instances with many objectives (e.g., 5 and 8 objectives cases), weakly Pareto-optimal boundaries (e.g., TN1-TN4 and mDTLZ3), linear $PF$s (e.g., TN1 and TN3), convex $PF$s (e.g., mDTLZ3), concave $PF$s (e.g., TN2, TN4, and DTLZ3), irregular $PF$s (e.g., TN3, TN4, DTLZ5, IMOP4, and IMOP6).

The following table summarizes the complete results on all the test problems by showing "Total +/=/-". "+", "=" or "-" denotes that the performance of the corresponding algorithm is statistically better than, similar to, or worse than that of BDNE based on Wilcoxon's rank sum test at 0.05 significant level. DTLZ, MP-DMP, and ML-DMP are not used in our experimental study, as the MaF test suite already covers them. We can find that our method still outperforms the other algorithms on most instances.

|Problem|# instances|EC-NSGA-II|DNPE|BDNE|
|:-------------:|:-----------:|:----------:|:--------:|:---------:|
|MaF1-MaF7|21|1/0/20|5/2/14| \ |
|MaF8-MaF9|6|0/0/6|1/0/5| \ |
|MaF10-MaF13|12|5/0/7|0/1/11| \ |
|mDTLZ1-mDTLZ4|12|0/0/12|1/2/9|\  |
|IMOP4-IMOP8|5|0/0/5|0/1/4| \ |
|Average rank||2.5893(3)|2.125(2)|1.2857(1)|


##
Best Regards,
Paper13080 Authors


## References
[1] Mesquita-Cunha et al. New $\epsilon$-constraint methods for multi-objective integer linear programming: A Pareto front representation approach. EJOR, 2023.

[2] Zhang et al. Hypervolume maximization: A geometric view of Pareto set learning. NeurIPS, 2023.

[3] Branke et al. Multiobjective optimization: Interactive and evolutionary approaches. 2008.

[4] Weerasuriy et al. Performance evaluation of population-based metaheuristic algorithms and decision-making for multi-objective optimization of building design. Building and Environment, 2021.

[5] Mena et al. Multi-objective two-stage stochastic unit commitment model for wind-integrated power systems: A compromise programming approach. International Journal of Electrical Power \& Energy Systems, 2023.

[6] Ekhtiari et al. Optimizing the dam site selection problem considering sustainability indicators and uncertainty: An integrated decision-making approach. Journal of Cleaner Production, 2023.

[7] Kirlik et al. Computing the nadir point for multiobjective discrete optimization problems. Journal of Global Optimization, 2015.

[8] Köksalan et al. Finding nadir points in multi-objective integer programs. Journal of Global Optimization, 2015.

[9] Özpeynirci et al. On nadir points of multiobjective integer programming problems. Journal of Global Optimization, 2017.

---

### Decision · Program_Chairs · 2024-09-25

**Decision:**

Accept (poster)

**Comment:**

This paper proposes a method for nadir objective vector estimation in multi-objective optimization based on the formulation of bilevel optimization. The experimental evaluation, including real-world problems, demonstrates the effectiveness and potential of the proposed method.

This paper treats an important problem and is well-written. The theoretical discussion is one of the strengths of this paper. All reviewers' scores are Accept or Weak Accept, and they recognize the novelty and effectiveness of the proposed method. Because the reviewers' opinions are consistent, I would recommend accepting this paper.